# Hygroscopicity of organic surrogate compounds from biomass burning and their effect on the efflorescence of ammonium sulfate in mixed aerosol particles

Ting Lei[1,2], Andreas Zuend[4], Yafang Cheng[2,3], Hang Su[2,3], Weigang Wang[1], Maofa Ge[1*]

[1]State Key Laboratory for Structural Chemistry of Unstable and Stable Species, CAS Research/Education Center for Excellence in Molecular Sciences, Institute of Chemistry, Chinese Academy of Sciences, Beijing, 100190, P. R. China

[2]Multiphase Department, Max Planck Institute for Chemistry, Mainz 55128, Germany

[3]Institute for Environmental and Climate Research, Jinan University, Guangzhou, China

[4]Department of Atmospheric and Oceanic Sciences, McGill University, Montreal, Quebec, Canada

*Correspondence to*: M. F. Ge (gemaofa@iccas.ac.cn)

**Abstract:** Hygroscopic growth factors of organic surrogate compounds representing biomass burning and mixed organic-inorganic aerosol particles exhibit variability during dehydration experiments depending on their chemical composition, which we observed using a hygroscopicity tandem differential mobility analyzer (HTDMA). We observed that levoglucosan and humic acid aerosol particles release water upon dehumidification in the range from 90 – 5 % relative humidity (RH). 4-Hydroxybenzoic acid aerosol particles, however, remain in the solid state both upon dehumidification or dehumidification and exhibit a small shrinking in size at higher RH compared to the dry size. For example, the measured growth factor of 4-hyroxybenzoic acid aerosol particles is ~0.96 at 90 % RH. The measurements were accompanied by RH-dependent thermodynamic equilibrium calculations using the AIOMFAC and the E-AIM models, the ZSR relation, and a fitted hygroscopicity expression. We observed several effects of organic components on the hygroscopicity behavior of mixtures containing ammonium sulfate (AS) in relation to the different mass fractions of organic compounds: (1) a shift of efflorescence relative humidity (ERH) of ammonium sulfate to higher RH due to the presence of 25 wt % levoglucosan in the mixture. (2) There is a distinct efflorescence transition at 25 % RH for mixtures consisting of 25 wt % of 4-hydroxybenzoic acid compared to the ERH at 35 % for organic-free AS particles. (3) There is indication for a liquid-to-solid phase transition of 4-hydroxybenzoic acid in the mixed particles during dehydration.

(4) A humic acid component shows no significant effect on the efflorescence of AS in mixed aerosol
particles. In addition, consideration of a composition-dependent degree of dissolution of crystallization
AS (solid-liquid equilibrium) in the AIOMFAC and E-AIM models leads to a relatively good agreement
between models and observed growth factors, as well as ERH of AS in the mixed system. The use of the
ZSR relation leads to good agreement with measured diameter growth factors of aerosol particles
containing humic acid and ammonium sulfate. Lastly, two distinct mixtures of organic surrogate
compounds, including levoglucosan, 4-hydroxybenzoic acid, and humic acid were used to represent the
average water-soluble organic carbon (WSOC) fractions observed during the wet and dry seasons in the
central Amazon Basin. A comparison of the organic fraction's hygroscopicity parameter for the simple
mixtures, e.g. $\kappa \approx 0.12$ to $0.15$ for the wet-season mixture in the 90 % to 40 % RH range, shows good
agreement with field data for the wet season in Amazon (WSOC $\kappa \approx 0.14\pm0.06$ at 90 % RH). This
suggests that laboratory-generated mixtures containing organic surrogate compounds and ammonium
sulfate can be used to mimic, in a simplified manner, the chemical composition of ambient aerosols from
the Amazon for the purpose of RH-dependent hygroscopicity studies.
**Introduction**
It is well established that biomass burning, as an important source of atmospheric aerosol particles, has
a wide range of climate effects that can be classified into direct radiative effects through light-absorbing
carbon aerosol particles and indirect effects by impact on cloud condensation nuclei and cloud
microphysics (Andreae and Gelencsér, 2006; Moosmüller et al., 2009; Hecobian et al., 2010; Rizzo et
al., 2011; Rose et al., 2011; Cheng et al., 2012; Engelhart et al., 2012; Lack et al., 2012; Jacobson, 2014;
Liu et al., 2014; Saleh et al., 2013, 2014). Atmospheric light-absorbing particles that arise from biomass
burning play an important role as a driver of global warming (Favez et al., 2009; Hegg et al., 2010; Lack
et al., 2012; Feng et al., 2013; Laborde et al., 2013; Srinivas and Sarin, 2013). According to the IPCC
report (Boucher and David, 2013), the climate forcing of black carbon aerosol particles may rival that of
methane, with a present-day global warming effect of up to 0.3–0.4 °C (Wang et al., 2014). Also, certain
types of aerosol particles emitted by biomass burning, when immersed into cloud droplets, absorb solar
radiation and facilitate water evaporation and cloud dispersion, which indicates an additional indirect
aerosol effect that counteracts the cooling effect of cloud droplets nucleated by aerosols (Powelson et al.,
2014). Therefore, a better understanding of the influence of aerosol particles from biomass burning on
cloud formation, precipitation, and Earth's radiative budget is required to comprehend biomass burning
aerosol properties and behavior.
The understanding of aerosol-cloud-climate impact of a vast range of biomass burning derived organic
compounds, however, is rather limited due to the complexity of biomass burning emissions, gas- and
aerosol-phase processing and the restricted availability of field measurements (Pratt et al., 2011; Lei et
al., 2014; Paglione et al., 2014; Srinivas and Sarin, 2014; Zhong and Jang, 2014; Ciarelli et al., 2015;
Arnold et al., 2015; Lawson et al., 2015; Gilman et al., 2015). Moreover, biomass burning particles are
often mixtures of water-soluble organic carbon, black carbon, varying amounts of inorganic components,
and water insoluble inclusions, such as mineral dust or poorly soluble organics (Väkevä et al., 2002;
Sadezky et al., 2005; Saarnio et al., 2010). An appreciable amount of organic compounds affect the
physicochemical properties of aerosols, such as hygroscopicity, liquid-solid and liquid-liquid phase
transitions, and chemical reactivity in liquid phases and/or on particle surfaces (Shiraiwa et al., 2013).
For example, equilibrium between the variable environmental water vapor mixing ratio and aerosol
particles may lead to substantial changes in particle size, and chemical composition, all of which can
influence light absorption and scattering (Seinfeld and Pandis, 2006; Zhang et al., 2016). RH-dependent
transitions between solid and liquid (aqueous) phases are also important in determining optical properties
(Martin et al., 2013; Wang et al., 2010; Kim et al., 2016; Wu et al., Denjean et al., 2015; 2016; Hodas et
al., 2015; Atkinson et al., 2015). Studies have shown that water-soluble organic matter from biomass
burning (approximately 70 % of total organic matter) can significantly suppress, enhance or have no
effect on the deliquescence (e.g. the RH at which deliquescence occurs at a certain temperature, the DRH)
and efflorescence process (e.g. the efflorescence RH, ERH) of present inorganic electrolytes. The effect
depends predominantly on the type of organics, mass fraction of organics relative to inorganic, and
particle size (You and Bertram, 2014; Zawadowicz et al., 2015; Hodas et al., 2015; Gupta et al., 2015).
Whole particles, individual phases within particles, or specific chemical compounds can undergo a range
of phase transitions including crystallization/efflorescence, dissolution/deliquescence, and liquid-liquid
phase separation as the relative humidity varies in the atmosphere. A number of laboratory studies have
focused on liquid-liquid phase separations within particles consisting of inorganic and organics fractions
(Svenningsson et al., 2006; Carrico et al., 2008; Dusek et al., 2011; Hodas et al., 2015). For example,
studies about liquid-liquid separation occurring in mixed organic-inorganic aerosols were performed by
Song et al. (2012, 2013) and You et al. (2013) using Raman and optical microscopy, establishing that
liquid-liquid phase separation occurs typically in mixed organics + ammonium sulfate particles with an
average elemental oxygen-to-carbon (O:C) ratio of the organic fraction of less than 0.6 and in some cases
for 0.6 < O:C < 0.8. You et al. (2013) further found that for a O:C ratio between 0.5 and 0.8, the
occurrence of liquid-liquid phase separation at moderate to high RH depends on the types of inorganic
salts present (i.e., the effective strength of the salting-out effect), e.g., $(NH_4)_2SO_4 \geq NH_4HSO_4 \geq NaCl \geq$
$NH_4NO_3$. Recently, the effect of a potential size-dependent morphology and dependence of the phase
separation mechanism on the organic/inorganic mass ratio in mixed aerosol was studied for mixtures of
poly- (ethylene glycol)-400 + ammonium sulfate using cryogenic-transmission electron microscopy
(Altaf et al., 2016). Therefore, many independent studies suggest that the occurrence of solid-liquid
and/or liquid-liquid phase separations, as well as related (temperature-dependent) RH levels of phase
transitions   (DRH, ERH, SRH), depend on the relative amounts of organic and inorganic aerosols
components and their non-ideal mixing behavior.
The expected physical state and morphology of aerosol particles containing mixtures of a wide range
of organic and inorganic salts/acids can, in principle, be predicted by a selection of specialized
thermodynamic equilibrium models. Such models include the Extended Aerosol Inorganic Model (E-
AIM) (Clegg and Seinfeld, 1998, 2006; available online: http://www.aim.env.uea.ac.uk/aim/aim.php),
the Aerosol Diameter Dependent Equilibrium Model (ADDEM) (Topping et al., 2004), the UNIversal
Quasichemical Functional Group Activity Coefficients model (UNIFAC) (Fredenslund et al.,1975;
Hansen et al., 1991), and the Aerosol Inorganic-Organic Mixtures Functional croups Activity
Coefficients model (AIOMFAC) (Zuend et al., 2008, 2011, 2012). These models have all been used to
predict atmospheric aerosol thermodynamic equilibrium for a variety of inorganic and organic systems,
yet not all of them can be used to compute non-ideal mixing in organic-inorganic systems. AIOMFAC
has been used to predict the distribution of components in multiple phases in a range of mixed organic-
inorganic systems and demonstrated its broad applicability in predicting liquid-liquid phase separation
in such mixtures (Zuend et al., 2010; Song et al., 2012; Zuend and Seinfeld, 2012; Shiraiwa et al., 2013;
Renbaum-Woff et al., 2016; Rastak et al., 2017).
Several previous experimental studies using the HTDMA technique (e.g. Zardini et al., 2008; Lei et
al.2014) show that the deliquescence of inorganic compounds is affected by the presence of organic
components, which manifests itself in a shift in the DRH of a salt compared to the corresponding organic-
free system. For instance, a clear shift of ammonium sulfate DRH was observed in the case of the
levoglucosan + ammonium sulfate system (Lei et al., 2014). Here we focus on investigating the
morphology, hygroscopicity and phase transitions of relevant organic compounds found in biomass
burning aerosol during the dehydration/dehumidification process. Moreover, we study how the presence
of organic compounds affects the water loss behavior of mixed organic-inorganic aerosols with
ammonium sulfate (AS) in the supersaturated state as well as after efflorescence of AS. In addition, we
compare the measured hygroscopicity behavior of mixed aerosol particles with predictions from the
Zdanovskii–Stokes–Robinson (ZSR) mixing rule, the E-AIM model and the AIOMFAC model.

**2 Methods**
**2.1 Aerosol system**
The three organic compounds levoglucosan, 4-hydroxybenzoic acid and humic acid were used as
surrogates for the rich-class of water-soluble organic components in biomass burning aerosols. The
influence of the distinct chemical structure of these compounds was studied with regard to the water
uptake and evaporation of the pure organic compounds as well as for mixed organic-AS-containing
particles. Furthermore, a comparison with field data from the Amazon was preformed to quantify the
ability of mixtures of these three organic compounds to mimicking the hygroscopic behavior of complex
ambient organic particles originating from biomass burning emissions. Here we focus on the
characterization of hygroscopic growth factors as well as solid-liquid and liquid-liquid phase transitions
during the dehumidification conditions. The chemical substances and their physical properties are
characterized in Table 1. All of the experimental solutions were prepared by dissolving in Milli-Q water
(resistivity $\geq$ 18.2 M$\Omega$) and the experiments were conducted at room temperature (~298 K). For the
mixtures of ammonium sulfate and organic surrogates the different mass ratios of AS: organic considered
are 3:1, 1:1 1:3. The chemical compositions of biomass-burning model mixtures are introduced in Table

141    2.

**2.2 Instrument design**
Figure 1 shows a schematic of the HTDMA instrument, more detailed information about this instrument's
setup, calibration and evaluation is described elsewhere (Lei et al., 2014; Jing et al., 2015, 2017; Liu et
al., 2015). Briefly, Poly-dispersed sub-micrometer aerosol particles are generated by atomizing (MSP
1500, MSP) a 0.05 weight % aqueous solution consisting of different mass fractions of inorganic, and
organic components, assuming that the composition of the formed aerosol particles is initially the same
as that of the solution used in the atomizer. Aerosol particles from an atomizer are routed though
homemade silica diffusion dryers and then pass through a Nafion gas dryer (Perma Pure Inc., USA).
After aerosol particles were dried to below 5 % RH (RH set point 1, RH1), they are directed to the
impactor; those aerosols with diameter less than 1 μm are allowed to pass it and subsequently pass
through a $^{85}$Kr electric charger to reach a near-Boltzmann distribution of charges (Liu et al., 1985). After
charging, the aerosol particles enter the first differential mobility analyzer (DMA1) at a sheath flow to
aerosol flow ratio of 4:0.3. The sheath flow is circulated by the diaphragm pump in the first loop DMA1
system, and its RH is kept constant at below 5 % RH. The resulting mono-disperse particle population,
selected within uncertainty by the DMA1, is then exposed to high RH conditions during which the aerosol
flow is humidified to 98 % RH by mixing water through a Nafion membrane humidifier at 30 °C. After
passing through a saturator (Perma Pure Inc., USA), the aerosols are dried to a target RH level (RH2)
through a series of two single-Nafion tubes (Perma Pure Inc., USA) with RH2 set to a value in the range
of 90 % to 5 % RH. Here, a pulse width modulator (PWM) circuit is used to regulate the sheath flow on
the basis of a proportional integral derivative (PID) system. When the second Nafion membranes allow
for regulating the sheath flow to a desired RH and for controlled flow into the sample stream until the
RH2 setting value is equal to the excess RH of sheath flow value (RH3), the mobility diameter of the
dehumidified of aerosols at target RH are measured with the second DMA (DMA2, a scanning DMA)
coupled with a condensation particle counter CPC (Model 1500, MSP). In addition, the residence time
between the humidifier and DMA2 is around 5 s, which is estimated to be sufficient for aerosols to grow
/shrink to equilibrium size at a certain RH setpoint. Also, due to recirculation of the sheath flow and the
pre-humidification of the aerosol flow, the sheath flow and aerosol sample flow are enabled to rapidly
reach the same RH.
**2.3 Theory and modeling methods**
Models were applied to explore the extent to which measured hygroscopic diameter growth factors
(HGFs), particle phase states, and phase compositions under sub-saturation conditions can be predicted
by thermodynamic equilibrium models. For the AS-containing systems studies, the current
thermodynamic equilibrium predictions account for a crystalline AS phase with solid-liquid equilibria
prior to the complete deliquescence of AS under hydration conditions. Similarly, the crystallization point
followed by a solid-liquid equilibrium of AS needs to be considered to predict the effect of organic
components in the mixed particles on the shift/suppression of AS efflorescence during aerosol
dehumidification, i.e. referring to processes occurring along the dehydration branch of a HTDMA
humidification-dehumidification cycle. The calculation of the ERH of AS in an organic-inorganic
solution is thermodynamically related to the solubility limit, but it is not strictly deterministic (unlike the
DRH) due to the stochastic nature of nucleation-and-growth of a crystal embryo. The molality of pure
AS at saturation in an aqueous solution is known, e.g. measured by Apelblat (1993) at 298.15 K as
$m_{AS}^{(sat)}$ = 5.790 mol/kg, while measurements are most often not available for the solubility limit of AS in
aqueous inorganic-organic systems. However, crystalline AS in equilibrium with an aqueous mixture
demonstrates a specific molal ion activity product (IAP) in that solution at a given temperature and
atmospheric pressure. For example, in the case of a ternary liquid mixture of levoglucosan + AS + water
in solid-liquid equilibrium (SLE) with a crystalline AS phase at a certain temperature T, a constant molal
ion activity product $IAP_{AS} = IAP_{AS}^{(sat)}(T)$ is established (necessary SLE condition). In this case the liquid
mixture is a so-called saturated solution with respect to AS. While the molar amount of AS in a saturated
solution depends on the other mixture constituents, the value of $IAP_{AS}^{(sat)}(T)$ is a function of temperature
only, since it is derived from the fixed chemical composition and associated chemical potential of the
crystalline phase. A reference value for $IAP_{AS}^{(sat)}(T)$ can therefore be calculated with the AIOMFAC
model from an experimentally determined solubility limit of AS in a known mixtures, such as the
molality of AS at the point of saturation in the binary aqueous system (water + AS). The RH at which
full dissolution of a solid phase upon humidification is just reached, the DRH, is directly related to the
conditions at which a saturated solution becomes subsaturated upon addition of water. Here the degree
of saturation with AS can be determined unambiguously by the computed value of $IAP_{AS}$ as a function
of mixture composition and temperature. Making use of these thermodynamic relationships, the
AIOMFAC-based equilibrium model is used to calculate the DRH and ERH of AS in the multicomponent
system, as outlined below. Detailed information on the modeling of solid-liquid equilibria and the IAP-
based prediction of ERH is given in Zuend et al. (2011) and Hodas et al. (2016). Briefly, the ERH is
determined based on the following equations:
$$\text{IAP}_{\text{AS}} = \left[a_{NH_4^+}^{(m)}\right]^2 \left[a_{SO_4^{2-}}^{(m)}\right]^1 \tag{1}$$
$$\text{IAP}_{\text{AS}}^{[\text{crit}]} = c_{\text{AS}} \times \text{IAP}_{\text{AS}}^{(\text{sat})} \tag{2}$$
Here $a_{NH_4^+}^{(m)}$ and $a_{SO_4^{2-}}^{(m)}$ are the molal activities of the ammonium and sulfate ions in solution (Zuend et
al., 2010). Molality basis is indicated by superscript "$(m)$" (which is not a mathematical exponent).
$\text{IAP}_{\text{AS}}^{(\text{sat})}$ denotes the molal ion activity product of AS at salt saturation computed by the thermodynamic
equilibrium model for any aqueous AS system at a certain temperature (here 298.15 K). The calculated
molal IAP at saturation of the corresponding binary salt solution is taken as the (known) reference value.
The RH at which this $\text{IAP}_{\text{AS}}^{[\text{sat}]}$ value is just reached in certain bulk solution at equilibrium with its
environment (in contrast to $\text{IAP}_{\text{AS}} < \text{IAP}_{\text{AS}}^{(\text{sat})}$ at higher RH), is the (bulk) DRH of AS. Similarly, the ERH
is determined at the point of crystallization by a critical IAP value denoted as $\text{IAP}_{\text{AS}}^{[\text{crit}]}$ (Hodas et al.,
2016), the value of $\text{IAP}_{\text{AS}}^{[\text{crit}]} > \text{IAP}_{\text{AS}}^{[\text{sat}]}$ expresses the need for reaching a critical IAP threshold (critical
level of AS super-saturation) for highly likely nucleation-and-growth of a new crystalline AS phase. The
multiplication factor $c_{\text{AS}}$ is used as a constant coefficient relating the IAP at AS saturation to the one
expected at the point of crystallization in aqueous mixed particles. From the comparison of laboratory
measurement of ERH for aqueous AS solution to the AIOMFAC-predicted $\text{IAP}_{\text{AS}}$ at that RH, the value
of $c_{\text{AS}} \approx 30$ was determined; this value is in particular applicable to submicron-sized AS droplets (Zardini
et al., 2008; Ciobannu et al., 2010).
An analogous approach is used for the ERH predictions with the E-AIM model, however, since E-AIM
provides activity coefficients and activities on mole fraction basis, denoted here by superscript "$(x)$"
(rather than molality basis), the value of $c_{\text{AS}}^{(x)}$ need to be determined separately for that model.
Expressing Eq. (1) by mole-fraction-based activities of $NH_4^+$ and $SO_4^{2-}$ and comparison to the
$\text{IAP}_{\text{AS}}^{(\text{sat},x)}$ and $\text{IAP}_{\text{AS}}^{(\text{crit},x)}$ computed by E-AIM for AS at the experimental solubility limit and ERH in
aqueous AS solutions, a value of $c_{\text{AS}}^{(x)} \approx 40$ was determined for the calculation with E-AIM.
As discussed by Lei et al. (2014), predicting of hygroscopic growth factors with E-AIM includes a
sophisticated composition-dependent solution density model, which considers the non-ideality effects on
apparent molar volumes used for the calculation of the solution density in mixed organic-inorganic
systems (Clegg and Wexler, 2011a, b). The AIOMFAC-based model applies a simpler solution density
treatment by assuming that the partial molar volumes of solution species are independent of non-ideal
interactions, i.e. the mixed solution density is calculated based on linear additivity of pure component
solid or liquid volume contributions to obtain the HGF at a given RH. Differences in the density models
are expected to lead to relatively small differences, typically on the order of the HTDMA measurement
error or less (e.g. Fig. 2a), in the application to HGF predictions, as demonstrated by Lei et al. (2014) for
the case of diameter vs. mass-based HGF of AS droplets. Both models include sophisticated sets of
equations to compute activity coefficients of all solution components in a thermodynamically consistent
manner.
## 2.4 $\kappa$-Köhler theory and computation of the hygroscopicity parameter $\kappa$
The hygroscopicity parameter, $\kappa$, is commonly used to characterize the relative hygroscopicities of
individual aerosol particles, known mixtures or complicated atmospheric aerosols (Petters and
Kreidenweis, 2007), and to model the composition-dependence of the solution water activity. The
saturation ratio, $S$, in the traditional Köhler equation (Eq. 3), over an aqueous droplet is calculated from
$$S = a_w \left( \frac{4\sigma_s M_w}{RT\rho_w D_{wet}} \right) \qquad (3)$$
Where $a_w$ is the mole-fraction-based water activity in solution, $M_w$ and $\rho_w$ are the molar mass of water
and the density of pure water in the liquid state at temperature $T$, respectively. $D_{wet}$, the "wet" particle
diameter at a given RH, is defined by $D_{wet} = HGF \times D_0$. $D_0$ denotes the diameter at dry conditions at
RH below 5 %. The solution surface tension is denoted by $\sigma_s$. In the "$\kappa$-Köhler theory", the bulk solution
water activity is described by a single parameter $\kappa$, with the hygroscopic parameter of the overall mixture
related to Eq. (3) by
$$\kappa_{HGF} = 1 - HGF^3 + \frac{HGF^3 - 1}{S} \exp\left[ \frac{4\sigma M_w}{RT\rho_w D_{wet}} \right] \qquad (4)$$
This expression describes effective values of $\kappa_{HGF}$ as a function of droplet diameter and $HGF$ at a
certain saturation ratio. In turn, known (measured) solution $\kappa_{HGF}$ values or component-specific $\kappa_i$
values can be used to parameterize or predict the $HGF$ curve of a mixture (Petters and Kreidenweis,

254      2007).

**2.5 GF data fit**
As described by Dick et al. (2001), the relationship between measured hygroscopic growth factors and
water activity can alternatively be parameterized by the following expression:
$\quad HGF = \left[1 + (c_1 + c_2 \times a_w + c_3 \times a_w^2)\frac{a_w}{1-a_w}\right]^{\frac{1}{3}}$ (5)
By substitution of Eq. (3) for $a_w$ in Eq. (5) and a fit to the measured HGF the three adjustable coefficients
$c_1$, $c_2$, $c_3$ of Eq. (5) were determined. The coefficient values are given in Table 3 for the different
organic compounds considered.
**2.6 GF prediction by ZSR**
The Zdanovskii-Stokes-Robinson mixing rule is widely used to approximate the water uptake of mixed
systems by assuming additivity of the water uptake of each individual component in the mixed particles
at a given RH (e.g., Malm and Kreidenweis, 1997). $HGF_{mix}$ is based on the $HGF_j$ of pure components $j$
and their corresponding volume fraction, $\varepsilon_j$ in the mixed particles.
$\quad HGF_{mix} = \left[\sum_j \varepsilon_j \left(HGF_j\right)^3\right]^{\frac{1}{3}}$ (6)

**3. Results and discussion**
**3.1 GF of single compounds systems**
Figure 2a shows the measured diameter growth factors of AS particles as a function of RH for both
humidification and dehumidification conditions. The measured ERH of 100 nm AS particles is
approximately 35 % RH at 298.15 K. The models predicted GF and predicted solid-liquid phase transition
of AS are in relatively good agreement with the experimental data and, in particular, the efflorescence
(crystallization) of AS is captured by the AIOMFAC and E-AIM models. The good model-measurement
agreement for the ERH is of course expected, since the aqueous AS system serves as the reference system
for determining the value pairs of $IAP_{AS}^{(sat)}$ and $c_{AS}$ on molality and mole fraction basis for use with
AIOMFAC and E-AIM, respectively (section 2.3). An ERH of 31 % to 40 % RH was reported by other
groups for a range of particle sizes and experimental techniques (Zardini et al., 2008; Ciobanu et al.,
2010). There are several factors that contributed to the variability of reported ERH values, such as
particles size, temperature, solution impurities and the stochastic nature of the homogeneous or
heterogeneous nucleation of a crystalline phase near ERH (Ciobanu et al., 2010).
In Fig. 2b, upon dehydration, no efflorescence of the levoglucosan aerosol particles is observed even
at RH below 10 %. The agreement of the HGF between the hydration and dehydration processes
demonstrates that these particles equilibrate with the surrounding water vapor under these moisture
conditions. For example, the measured diameter growth factors of levoglucosan particles at 80, 60, and
30 % RH are 1.19, 1.09, and 1.03, respectively, which are similar to results obtained for the hydration
process of such particles. Levoglucosan has a DRH of ~80 % to 83 % (for a bulk system) at 293 to 298 K
(Mochida and Kawamura, 2004; Zamora et al., 2011). The similarity of diameter growth factors both
under hydration and dehydration conditions even below the DRH of levoglucosan is explained by the
lack of crystallization of levoglucosan upon drying to low RH and the presence of a metastable
supersaturated aqueous levoglucosan solution in both the hydration and dehydration modes for
experiments initiated with liquid solution droplets (Mochida and Kawamura, 2004; Chan et al., 2005;
Svenningsson et al., 2006). A possible reason for a persistent metastable supersaturated solution states is
that levoglucosan particles remain liquid (possibly a viscous liquid state) upon drying to below 5 % RH,
which was also observed previously with a reported ERH < 4% RH (Mochida and Kawamura, 2004;
Chan et al., 2005). Also, the measured diameter growth factors of levoglucosan particles are in good
agreement with these estimated from the standard UNIFAC model within the E-AIM model and the
AIOMFAC model, within experimental uncertainty. The UNIFAC models within E-AIM and AIOMFAC
are based on the original model expressions by Fredenslund et al. (1975) and both include the extensive
parameter set by Hansen et al. (1991) as well as revised parameters for certain group interactions of water
with carboxyl and hydroxyl groups by Peng et al. (2001). Of relevance for levoglucosan and other sugar-
like compounds, the AIOMFAC model also contains certain revised group parameters for hydroxyl
groups and special alkyl groups for their interactions with water, introduced by Marcolli and Peter (2005)
for polyols, as further detailed in Zuend et al. (2011). However, the molecular structure of levoglucosan
with several polar functional groups in close vicinity may account for a small deviation between models
and measured *HGFs* at RH below 70 %, because intramolecular interactions are not fully considered by
these models.
The measured diameter growth factors of 4-hydroxybenzoic acid particles shown in Fig. 2c
demonstrate untypical increase in diameter of 4-hydroxybenzoic acid particles during dehumidification
from 90 to 10 % RH, which is consistent with previous diameter growth factor for a few solid particles
(Mochida and Kawamura, 2004). The organic particles measured are likely always in the effloresced, i.e.
crystalline state apparently even at high RH. The apparent increase in diameter during dehumidification
may be explained by particle shape restructuring, since the (poly)-crystalline particles are likely non-
spherical at dry conditions, but may become more sphere-like in shape when exposed to higher RH
(Mikhailov et al., 2004). Also, no ERH of 4-hydroxybenzoic acid in the dehydration mode was observed
during the experiments; the likely reason is that the highest RH reached in the humidifier was
approximately 98 %, which may be below the ERH of 4-hydroxybenzoic acid, reported as above 98 %
RH in another study (Mochida and Kawamura, 2004). As discussed previously by Lei et al. (2014), our
HTDMA experiments are carried out such that RH = 98 % is reached initially before dehumidification
to a series of relative humidities at set point RH2 (90 % – 5 % RH), the crystallization of the organic,
however, could occur at above 90 % RH. In addition, deviations between measurements and model
prediction are obvious in Fig. 2c. The observations surpass by far the expected error in model
performance, which is typically less than 0.05 in HGF units for RH < 85 %, as indicated also by an
intercomparison of the AIOMFAC and E-AIM predictions in Fig. 2c and much improved model-
measurement agreement for the case of mixed 4-hydroxybenzoic acid + AS particles shown in Fig. 4
(discussed in Section 3. 2. 2). However, note that the validity of the shown model predictions in Fig. 2c
depends on whether the assumption of a liquid solution droplet is plausible. Therefore, it is no surprise
that the model-predicted curves deviate from the experimental hygroscopic behavior of 4-
hydroxybenzoic acid particles. Morphology effects, such as the restructuring of non-spherical
polycrystalline particles over a certain RH range or liquid-liquid phase-separated particles of non-
spherical shapes, have been discussed by several groups (Sjogren et al., 2007; Reid et al., 2011; Lei et
al., 2014). In the case of hygroscopic growth and deliquescence under hydration conditions for 4-
hydroxybenzoic acid particles and mixtures of 4-hydroxybenzoic acid with ammonium sulfate. An offset
between measurement and model predictions was observed both in the RH range below the deliquescence
of the particles and above it, i.e. above 80 % RH, (Lei et al., 2014). It is suggested that deviations are
primarily caused by a change in solid-state particle morphology during hydration, leading to a
restructuring of the polycrystalline particle shape towards more compact, near-spherical shape as the RH
approaches the particle deliquescence point. This would explain rather uncommon HGF values of less
than 1.0 at elevated RH also shown in Fig. 2c. Similar behavior was found for experimental growth
factors of mixtures containing adipic acid and AS and systematic deviations between the associated ZSR
predictions and observations by Sjogren et al. (2007). Thus, while experimental data hint to the possible
influence of non-spherical particles and their humidity-induced restructuring as a source of uncertainty,
model predictions of HGF, such as those with the AIOMFAC model, assume by default a spherical
particle shape even for solid phases and/or in cases where LLPS is present.
The measured HGF curves of humic acid aerosol particles during dehumidification and humidification
measurements do not agree very well within experimental uncertainty, in particular above 70 % RH. For
instance, the growth factor of humic acid aerosol particles at 80 % RH is 1.2 according to the
dehumidification measurement, which is higher than hygroscopic growth factor of humic acid particles
in the humidification mode at the same RH. Humic acid aerosol particles shrink continuously due to loss
of water content in the range from 90 % to 10 % RH. For example, a stepwise change in the water
absorption and desorption behavior within different RH range was observed in the case of Nordic Aquatic
Fulvic Acid (NAFA) and Suwannee River Fulvic Acid (SRFA) by Chan and Chan (2005). These
hygroscopic behaviors suggest that humic acid particles and structurally similar compounds remain some
water down to the low RH levels achieved in the instruments (imperfect drying during particle residence
in the instrument). In addition, the experimental growth factor of humic acid aerosol particles during
dehumidification can be represented well by fitting Eq. (5) to the measurements. The determined fit
parameters are listed in Table 3. The humic acid sample used (Aldrich, 99%) are a mixture of different
poly-carboxylic acids of undefined chemical structure. However, specific information on the chemical
structure and mixture composition is necessary for corresponding model predictions with AIOMFAC and
E-AIM. Therefore, no such model calculations are shown in Fig. 2d.
**3.2 GF of mixtures of organic surrogate compounds + ammonium sulfate**
Biomass burning aerosol particles are likely mixtures of a diversity of inorganic constituents and organic
compounds in the atmosphere. For example, particles may consist of a combination of ammonium sulfate
mixed with low- and semi-volatile organics from biomass burning emissions (Lee et al., 2003; Zhang et
al., 2007; Pratt and Prather, 2010). Different water solubilities, and hygroscopic behavior of distinct
organic compounds may affect the hygroscopic growth factors of mixtures of partially or fully dissolved
inorganic and organic components. For example, Bodsworth et al. (2010) studied the effect of different
mass fractions of citric acid on the efflorescence properties of mixed citric acid-ammonium sulfate
particles at lower temperature and concluded that adding citric acid decreases the ERH of ammonium
sulfate in the mixed aerosol particles. These hygroscopic behaviors of mixed aerosol particles, including
phase transition in the range from moderate to low RH, are the focus of attention in this study.
**3.2.1 Mixed system: levoglucosan + ammonium sulfate**
Figure 3 shows measured growth factors of mixed aerosol particles containing levoglucosan +
ammonium sulfate with different dry state organic-to-inorganic mass ratios (1:3, 1:1, 3:1) in the RH range
from 90 % to 10 %. There is a reduction in the diameter growth factor of aerosol particles containing
levoglucosan and AS with increasing levoglucosan mass fraction, as expected from a ZSR-like additivity
concept of hygroscopicity. When the concentration of levoglucosan is low (25 wt %), a clear
efflorescence signature of AS found, within the ERH shifting to a higher RH (40 % - 45 %) in comparison
to the ERH of pure AS occurring at 33 %-35 % RH (Fig.3a). A similar phenomenon has been found for
the certain mixtures of NaCl and Nordic Aquatic Fulvic Acid (NAFA), in which the crystallization of
NaCl shifted to higher RH by mixing with NAFA at a mass ratio of 1:1 (Chan and Chan. 2003). With
increasing mass fraction of levoglucosan (i.e., 50 wt % and 75 wt %), the mixtures release water gradually
and no crystallization of AS was observed. Although a small step in the growth factor curve might have
occurred (indicative of the crystallization of AS), it cannot be detected with sufficient certainty by our
measurement setup. The rather high viscosity of solutions containing levoglucosan is expected to
increase considerably toward RH (Marshall et al., 2016). This increase in viscosity might impede the
crystallization of AS in the mixed systems on the time scale of the experiment. Mass transfer limitation
effects on the deliquescence or efflorescence process of crystalline organic particles and the water uptake
or evaporation have been investigated in several experimental studies (Peng et al., 2001; Choi and Chan,
2002; Chan and Chan, 2005; Sjogren et al., 2007; Zardini et al., 2008; Ciobanu et al., 2010; Smith et al.,
2012; Mikhailov et al., 2013; Hodas et al., 2015). Mass transfer limitations may impact the outcome of
experiments significantly if the characteristic time scales for equilibration is similar to or larger than the
residence time of particles in the experimental setup. In this study, the total residence time of the aerosol
sample during the equilibration phase before entering the DMA2 is about 8 s. In order to improve the
probability that the particles reach equilibrium with the target RH during this residence time, the
monodisperse aerosol selected by DMA1 is first humidified to 98 % RH. The aerosol particles are then
exposed to a lower target RH by a two-step process using double Nafion tubes. Kerminen (1997)
estimated the necessary residence time for achievement of water equilibrium of aqueous droplets to be
between 8 ×10-6 s and 0.1 s for 100 nm and 500 nm particles, respectively. Therefore, the typical
residence time of a few seconds in the humidification or dehumidification section in a HTDMA
measurement is assumed to be sufficient for most equilibrium hygroscopicity measurements (Brooks et
al., 2004; Mikhailov et al., 2004). Moreover, our HGF results for the three pure organic components are
in good agreement with respective data by Mochida and Kawamura, (2004), Brooks et al., (2004) and
Chan and Chan (2005) conducted with different techniques and/or residence times. However, there are
cases where water equilibration could be impeded substantially in the presence of highly viscous or
glassy particles at low RH, e.g. for ternary sucrose + NaCl + water particles of > 6 μm in diameter studied
by Bones et al. (2012), who report an equilibration time scale > 1000 s for such particles. Note that, aside
from viscosity, there is an important size-dependence of the particles on the equilibration time scale (e.g.
Koop et al. 2011). For aqueous 100 nm particles used in HTDMA experiments at room temperature,
Bones et al. (2012) indicate that the equilibration time scale for water is likely only of concern for RH <
10 % in such an instrument. We therefore conclude that the residence time of 8 s is very likely sufficient
to allow for equilibrium HGF measurements in dehydration mode, at least down to 10 % RH (when
starting with aqueous solution droplets).
Mass transfer effects in hygroscopicity measurements of aerosol particles during hydration conditions
have been encountered previously, particularly when a solid-liquid phase transition (deliquescence) is
involved (Peng et al., 2001; Chan and Chan, 2005; Sjogren et al., 2006). For example, Peng et al. (2001)
observed in electrodynamic balance (EDB) experiments under conditions of very slow humidification
that glutaric acid aerosol particles showed a deliquescence phase transition in the RH range from 83 to
85 % over the course of several hours. This is a much longer time span than that of ~ 40 min for the
deliquescence of other super-micron sized dicarboxylic acid particles (e.g., malonic acid) in EDB
experiments. This observation indicates that the solid-liquid phase transition of glutaric acid particles
may likely be mass transfer limited during the hydration process. In this context, it is possible that the
deliquescence of initially solid, pure 4-hydroxybenzoic acid particles at RH > 97 % is further impeded
by slow dissolution, which could have led to the absence of deliquesced particles (Fig. 2c) on
experimental time scale.

427       In addition, the measured diameter growth factors of mixtures of levoglucosan and AS are compared

to calculations of hygroscopic growth by the E-AIM and AIOMFAC models. The E-AIM prediction is
in relatively good agreement with results from the HTDMA measurement but typically overestimates the
water content of particles consisting of organic-AS mixtures at the RH range close to the ERH of AS.
The liquid-solid phase transition of ammonium sulfate in the mixed particles is considered in the E-AIM
assumptions as described in Section 2.3. There is a more distinct shift of ERH of AS with higher mass
fractions of levoglucosan. In the case of the AIOMFAC and E-AIM model predictions, it is assumed that
the diameter growth factor contribution from AS is zero below the predicted ERH, i,e. there the growth
factor deviation from 1.0 is solely due to the organic water uptake. The model prediction shows a slight
deviation from the measurements, which may be in part due to (i) model uncertainty in the correct
description of the hygroscopicity of levoglucosan, (ii) due to incomplete representation of AS +
levoglucosan interactions in aqueous solutions and (iii) in part due to measurement error. Also, in the
case of mixtures consisting of AS and levoglucosan with organic-to-inorganic dry mass ratio of 3:1 (75
wt % levoglucosan of dry particle composition), the underestimation of the growth factor by the
AIOMFAC model at RH < 35 % in comparison to the measurements is explained in part by the model
prediction of AS efflorescence (which seems to be absent in the measurements). However, with
decreasing AS mass fraction, the hygroscopic behavior of levoglucosan dominates the diameter growth
factors of the mixtures, in relative agreement with the AIOMFAC-modeled "dehydration branch"
prediction. Minor differences in the AIOMFAC prediction vs. -measurement for diameter growth factors
of mixed levoglucosan and AS in the RH range of 35 - 25 % here might be attributed to mixture viscosity
effects at the higher levoglucosan contents, which may suppress the efflorescence of AS in the mixed
systems on experimental timescale or it could simply be due to sufficient miscibility of dissolved AS in
the aqueous levoglucosan solution (beyond that predicted by the model), such that a small step-change
due to AS efflorescence could be beyond the experimental detection range. As a result, accounting for
the effect of the organic components on the diameter growth factors of mixtures within aerosol particles
is crucial to model accurately the equilibrium hygroscopic behavior.
**3.2.2 Mixed system: 4-hydroxybenzoic acid + ammonium sulfate**
Mixtures of 4-hydroxybenzoic acid + AS with different organic mass fraction (25, 50, 75 wt %) exhibit
a gradual water desorption before the AS fraction of the particle effloresces at a certain RH. With
increasing 4-hydroxybenzoic acid mass fraction, no discontinuity step at the corresponding ERH in the
dehydration curve of mixtures is observed. This suggests the presence of 4-hydroxybenzoic acid in the
liquid state retards or offsets the efflorescence of AS in the mixtures. An interesting, yet contrasting
phenomenon was observed for the hydration process of aerosol mixtures containing 4-hydroxybenzoic
acid and AS by Lei et al. (2014). For the case of these mixtures during moistening, the deliquescence of
ammonium sulfate in the mixed particles remains unaffected, within experimental resolution, by the
presence of 4-hydroxybenzoic acid (Lei et al., 2014). Similar behavior has been observed for particles
containing certain organic acids of limited water-solubility mixed ammonium sulfate (Choi and Chan,
2002; Chan and Chan, 2003). For example, mixtures for succinic acid + ammonium sulfate showed no
substantial influence on the deliquescence RH of ammonium sulfate in the hydration process (Choi and
Chan, 2002). However, a clear RH shift of the deliquescence phase transition of ammonium sulfate or
sodium chloride was determined for mixed particles containing organic acids of higher water-solubility
and O: C ratio, such as citric acid and malonic acid (e.g. Choi and Chan, 2002). The DRH and ERH of
pure organics and AS in the mixed organic-AS particles are summarized in Table 4, the measurements
indicate that 4-hydroxybenzoic acid has a significant effect on the efflorescence of AS when present in
sufficient amount. Also, there is a clear reduction in the diameter growth factors prior to crystallization
for mixtures with increasing 4-hydroxybenzoic acid mass fraction.

473        The measurements of mixtures consisting of 4-hydroxybenzoic acid and AS are compared with model

predictions based on different assumptions about the phase state of the organic component, since the
deviation from measurements might partly be explained by a transition in the physical state of the organic
component. The E-AIM model prediction is referring to a system where the mixtures of 4-
hydroxybenzoic acid is assumed to be in the liquid state at all RH levels, which the efflorescence of AS
is considered. Neglecting the potential efflorescence of the organic component in the dehydration branch
makes a systematic offset more obvious prior to the efflorescence of AS. A good E-AIM model-
measurement agreement occurs below the predicted ERH of AS for mixed particles. The overestimation
of HGFs before the efflorescence of AS is explained by the AIOMFAC model prediction with distinct
assumptions about the organic phase state. A possible reason for the departure of model-measurement
agreement at RH < 80 % is that there are two liquid-to-solid phase transitions, occurring in the mixed
particles: a gradual one for the organic component and a step-like one for AS at lower RH. This
phenomenon is shown in the grid square range in Fig. 4 and supported by comparison of the measured
HGF data with AIOMFAC-based predictions for two assumptions about the organic phase state,
especially in the case of mixtures with 50 and 75 wt-% organic. We acknowledge that the model
predictions of the HGF curves for the two organic phase state assumptions differ within experimental
error for the case shown in Fig. 4a, indicating that alternative explanations, such as model/measurement
uncertainty in the absence of a liquid-solid phase transition could explain the observations. In the Fig.
4b, good agreement between measurements and the AIOMFAC model prediction with liquid organic
assumption is found for RH > 65 %, while for RH ≤ 60 % the experimental data agree very well with the
dashed red model curve for the case with consideration of a solid organic component. It suggests that
crystallization followed by gradually increasing partitioning of organic from the solution to the solid
organic phase occurs in the range from 70 % to 60 % RH under conditions of dehumidification. Similarly,
a liquid-to-solid phase transition occurs for the organic: AS mass ratio of 3:1 cases in the range from 80 %
to 50 % RH. Meanwhile, AS remains dissolved in a supersaturated aqueous solution phase. Moreover,
the AIOMFAC-based equilibrium model predicts a liquid-liquid phase separation (LLPS) to occur at RH
below ~ 90 % for the calculation cases with the assumption of the organic in the liquid state (for all three
organic mass fractions in Fig. 4). This prediction leads to a liquid phase enriched in 4-hydroxybenzoic
acid with some water and AS dissolved and a coexisting liquid phase enriched in AS and water. The onset
of the LLPS during dehumidification leads to the kink in the red model curve near 90 % RH, since the
slope of the HGF curve with RH changes in a non-smooth manner at the point of the LLPS phase
transition. This change in slope is not noticeable from the experimental data alone, but the model-
measurement comparison for the range above 80 % RH shows very good agreement. The two liquid
phases will likely remain separated until nucleation of a crystalline 4-hydroxybenzoic acid phase occurs
followed by gradual partitioning of the organic acid to the solid phase with decreasing RH (to ~50 %
RH), at which point only a single liquid phase (an aqueous AS phase with a tiny amount of dissolved
HA) will remain until efflorescence of AS occurs. Above ~90 % RH, a single, homogeneous liquid phase
is predicted to exist. Interestingly, this AIOMFAC model-measurement comparison (Fig. 4, especially
panels b and c) provides reasonable evidence that 4-hydroxybenzoic acid remains dissolved and therefore
in a liquid phase state at high RH in the mixed particles upon dehumidification (it is present in both liquid
phases below 90 % RH, but highly enriched in the AS-poor phase). In contrast, in the case of pure 4-
hydroxybenzoic acid aerosol particles, particles exposed to initial RH of $\geq$ 90 % remain in the solid state
( or crystallize at RH > 90 %) in the dehydration mode (Fig. 2c). What factors contribute to keeping the
organic in the liquid solution? It is possible that the aerosols generated with those mixed solutions were
allowing the 4-hydroxybenzoic acid to fully dissolve as the AS provides substantial particle phase water
content (within short time) into which the organic can be dissolved and may then further contribute to
water uptake associated with the organic's hygroscopicity (unlike in the case of the pure 4-
hydroxybenzoic acid particles). The 4-hydroxybenzoic acid remains dissolved in the mixture, possibly
supersaturated with respect to the crystalline organic state (similarly to how AS stays supersaturated at
RH below the DRH during drying). We consider this a reasonable explanation for the observed HGF data
from the HTDMA in comparison to the different AIOMFAC-based curves.
**3.2.3 Mixed system: humic acid + ammonium sulfate**
Figure 5 shows that the experimental diameter growth factors of mixtures consisting of humic acid (HA)
and AS with dry mass ratios of 1:3, 1:1 or 3:1 decreases with increasing mass fraction of HA at RH >
35 %. For example, at 35 % RH the measured HGF are 1.1, 1.05, 1.05 for the particles consisting of 25
wt %, 50 wt % and 75 wt % humic acid. In comparison, the diameter growth factor of pure supersaturated
AS particles is ~ 1.13 just prior to efflorescence of AS. Humic acid, unlike levoglucosan and 4-
hydroxybenzoic acid aerosol particles, has no noticeable effect on the efflorescence point of AS in the
mixed aerosol particles. Results of the ZSR model agrees well with measured hygroscopic growth for
mixtures within the experimental error. The ZSR curves shown in Fig. 5 are based on the RH-dependent
fitted hygroscopic growth factors of humic acid with Eq. (5) and the AIOMFAC predicted diameter
growth factors of AS in the dehydration mode. The success of the ZSR mixing rule for this system
suggests that interactions of organic molecules with ammonium sulfate ions in aqueous solution will only
marginally affect the hygroscopic growth factors of the mixtures containing humic acid and AS. Due to
the lack of detailed information about the actual chemical structures of humic acid samples used, it was
not possible to perform E-AIM and AIOMFAC model predictions for comparison with the measurement.
**3.3 Mixtures of biomass burning organic surrogate components with ammonium sulfate**
According to Decesari et al, (2006), sampling of aerosol particles, including the WSOC fraction, was
conducted from September 9 to November 14, 2002 in their field study, the sampling time was subdivided
into different periods. Despite of significant changes in the chemical composition of tracer compounds
from the dry to the wet period, the functional groups and general chemical classes of WSOC changed
only to a small extent in the Amazon basin near Rondônia, Brazil. Model compounds represent semi-
quantitatively (presence/abundance of functional groups) and the chemical structure of WSOC can be
used as surrogates in microphysical models involving organic aerosol particles over tropical areas
affected by biomass burning scenarios (Andreae et al., 2002; Artaxo et al., 2002; Rissler et al., 2006;
Decesari et al., 2006). Here, we focus on experimental observations and model calculations for relatively
simple mixtures of inorganic-organic surrogate components reflecting mixtures of aerosol components
found during different seasons during biomass burning events. However, we are fully aware of that fact
that actual biomass burning aerosols are typically much more complex in terms of particle chemical
composition. Aerosol particle properties from biomass burning events depend on the types of sources,
external/internal population mixing state, water-solubilities, and phase state of the diversity of organic
compounds and their mixing with inorganic constituents during different time periods in the field (e.g.
Decesari et al., 2006).

**3.3.1 Mixtures system: mix-bio-dry and mix-bio-wet aerosol particles**

Figure 6a shows the observed small differences in the hygroscopicity parameter κ for mixtures of organic
surrogate components and ammonium sulfate representing biomass burning particles during the dry and
wet periods in the Amazon, respectively. Hygroscopicity parameter values for bio-mix-dry aerosol
particles were determined to be between 0.16 and 0.18 with decreasing RH in the range from 90 % to
40 % RH. The $\kappa$ value representing the wet period in the Amazon is shown in Fig. 6b, derived from
laboratory HTDMA measurements in the range from 90 % to 40 % RH. A similar trend of an increase in
$\kappa$ with a decrease in RH has also been observed by Cheung et al. (2015). Their observation is based on
ambient particle measurement with a HTDMA in Hong Kong, therefore probing particles of more
complex compositions in the field campaign. The variability of the hygroscopicity parameter in sub-
saturated conditions reveals some limitations of a single-parameter hygroscopicity model for applications
over a wide range of RH. At low, intermediate and high RH levels, differing degrees of solution non-
ideality, potential for liquid-liquid phase separation, water-solubility limitations of organics in ambient
organic-inorganic particles, and assumptions about constant/variable surface tension may all play a role
(Mikhailov et al., 2009; Wex et al., 2009, Rastak et al., 2017; Ovadnevaite et al. 2017; Wang et al., 2017).
In the case of κ of organic surrogates mixed with ammonium sulfate, the relevant κ value range is ∼ 0.12
to 0.15 obtained from 90 % to 40 % RH. The measured $\kappa$ values of the mixtures are compared to field
data of HTDMA and CCN measurements conducted at a remote rainforest site in the central Amazon
during the dry and wet seasons (Whitehead et al., 2016; Pöhlker et al., 2016), which are consistent with
$\kappa$ obtained at similar field sites (within the uncertainty of experiments). The likely reason for a relatively
good agreement between the hygroscopicity of the laboratory mixtures and the field data is that the
organic mass fractions of the mix-bio-dry and mix-bio-wet mixtures are chosen in our laboratory
experiments to be similar to those of the latest field data from Amazon. For example, Pöhlker et al. (2016)
obtained the effective hygroscopicity parameters $\kappa$ between $0.3 \pm 0.01$ and $0.15 \pm 0.01$ based on the
organic mass fraction range from 0.65 to 0.97 in the dry season by aerosol chemical speciation monitor
(ACSM) and CCN measurements. The predicted $\kappa$ values of the mixtures at various RH levels shown in
Fig. 6 (black curves) are obtained by application of Eq. (4) with use of the RH-dependent fitted HGFs of
the organic surrogates (Eq. 5), the predicted growth factor of AS by the AIOMFAC model (for the
humidification case) and the volume fraction based mixing rule for a mixture's HGF (Eq. 6). For these
calculations, a solution surface tension of 0.072 J m$^{-2}$ was assumed. These predictions agree relatively
well with the experimental $\kappa_{dry}$ and $\kappa_{wet}$ values obtained from the HTDMA over a wide range in RH
referring to dehumidification conditions (no solid AS). Furthermore, the combined approach of Eqs. (4-
6) allows for a prediction of the change in $\kappa$ at high RH towards water vapor super-saturation. A small
difference in $\kappa$ between sub- and super-saturated conditions is observed for our mixed systems when
comparing the HTDMA data and predictions at 90 % RH with the predictions near 100 % RH and the $\kappa$
values from the CNN field measurements. The difference is more pronounced for the wet season case.
Rastak et al. (2017) observed a marked difference in apparent hygroscopicity and related mixture $\kappa$ of
the organic aerosols (AS-free) occurring in the case of monoterpene-derived secondary organic aerosol
(SOA) for sub- vs. super-saturated conditions. A smaller difference was reported for the isoprene-derived
SOA (Pajunoja et al., 2015: Rastak et al., 2017), more like the difference observed here for the mixtures
containing AS (and therefore having overall higher $\kappa$ values than typical salt-free organic aerosols).
Rastak et al. (2017) attribute the distinct difference in $\kappa_{SOA}$ of the monoterpene SOA to the limited mutual
solubility of certain SOA components in water, because a single liquid organic-phase of monoterpene
oxidation products is present at RH below 95 %, but over a RH range above 95 %, liquid-liquid phase
separation is observed by optical microscopy as well as predicted by the AIOMFAC-based equilibrium
model. In the mix-bio-wet and mix-bio-dry cases shown in Fig. 6, the likely reason for the change in
characteristic mixture hygroscopicity is not necessarily due to a liquid-liquid phase separation at high
RH. For example, the $\kappa$ parameter obtained from field data is ~0.15±0.06 at 90 % RH, while its value
reaches ~0.18±0.04 at RH > 100 % (just prior to CCN activation). A likely reason for the difference is
that hygroscopic particles, especially those containing sparingly soluble organics like 4-hydroxybenzoic
acid, take up water dramatically above 95 % RH when approaching 100 % RH (Huff Hartz et al., 2006;
Chan et al., 2008; Rastak et al., 2017), which is clear from model predictions, as demonstrated in Fig. 6
by application of Eq. (4). The predicted curve in the mixture's effective $\kappa$ parameter may well capture
the change in hygroscopicity under such high RH conditions. Consequently, for a precise representation
of the hygroscopic growth behavior (e.g. HGF) at high RH (> 95 %) by the $\kappa$-Köhler model, the value of
$\kappa$ would need to be varied. While a variable $\kappa$ value is contrary to the attempted simplicity of the single-
parameter $\kappa$-Köhler model, it is at least advised to consider that $\kappa$ values derived from HGF data at 80 %
or 90 % RH may not apply accurately for the calculation of CCN activation properties of such biomass
burning aerosols.
To summarize, there is small difference in hygroscopicity parameters between sub-saturated
measurement conditions at 90 % RH in the laboratory with HTDMA and supersaturated conditions using
CCN measurements, in agreement with the findings of other studies. At regional scale, in the dry and wet
period, the hygroscopic behavior in some extent of the Amazon rainforest is influenced significantly by
the biomass burning emissions, which enhances CCN activity and droplet number concentrations in
warm clouds in that region and influences the radiation balance and cloud life time (Pöschl et al., 2010).
Underestimation of organic surrogate component mass fractions in the mixed particles or organic:sulfate
mass ratios may be responsible for the slight differences in the determined $\kappa$ parameters of the laboratory
and field measurements.

**4. Conclusions**
A number of field-based hygroscopicity studies about biomass burning aerosol focus on the growth
factors of mixtures at high RH (e.g. 90 % RH). However, less attention has been paid to the growth
behavior at low to moderate RH, limiting the database for accurate estimates of particles optical and
radiative properties over those lower RH ranges. However, this is a RH range in which water uptake or
release behavior demonstrates a considerable variability among different organic-inorganic systems. The
occurrence or suppression of a liquid-solid phase transition affects the physicochemical particle
properties in a relative narrow RH range, potentially leading to particles of different morphology and
physical states, affecting effective particle size and density. In this work, measurements and
thermodynamic equilibrium predictions for organic-inorganic aerosols related to components from
biomass burning emissions demonstrate a diversity of hygroscopic growth/shrinking behavior. For
example, in the case of aerosol mixtures containing levoglucosan and ammonium sulfate, the presence
of levoglucosan may cause the efflorescence of AS to occur at higher RH than in pure aqueous AS
particles-or it may completely suppress AS efflorescence, as observed for mixtures with a high
levoglucosan mass fraction. The growth curves predicted with an AIOMFAC-based thermodynamic
equilibrium model reproduce the observations in most cases reasonably well and we demonstrate the
usefulness of predictions with different assumptions about the physical state of the organic components
for the interpretation of experimental data, such as in the case of mixtures of 4-hydroxybenzoic acid and
ammonium sulfate. However, the accurate prediction of AS efflorescence or its suppression in mixed
particles is difficult. The E-AIM-predicted growth curves reproduce the measured hygroscopic behavior
relatively well for the consideration of the effect of 4-hydroxybenzoic acid on the hygroscopic behavior
of mixtures with ammonium sulfate, which leads to suppression of the ammonium sulfate efflorescence.
In the case of mixtures of humic acid and ammonium sulfate, continuous water desorption of aerosol
particles shows no significant effect on the efflorescence of ammonium sulfate. Also, as expected, there
is a clear reduction in the diameter growth factor of mixed systems, in comparison with that of pure AS
particles. In addition, the small difference of hygroscopicity parameters of mix-bio-dry and mix-bio-wet
systems between measured data in the laboratory using HTDMA and the field using CCN activity
measurements is due to the similar O:C ratios of organic surrogate compounds and ammonium sulfate
mass fractions used in the model mixtures when experimental $\kappa$ data from sub- and super-saturated water
vapor conditions are compared.
The range of measurement-model comparisons presented in this study indicate that providing accurate
thermodynamic model predictions of the hygroscopic growth behavior of mixed organic-inorganic
systems remains a challenging problem. At moderate and low RH, where aerosol solution phases become
highly concentrated, step-like or gradual crystallization and related solid-liquid equilibria may occur with
high sensitivity to the organic/inorganic mass ratio and the chemical nature of the mixture constituents.
To further improve thermodynamic equilibrium models for the prediction of hygroscopicity and phase
transitions, controlled laboratory experiments with single solutes and/or with mixed organic-inorganic
systems of known phase state will be useful to constrain model parameters. Ideally, such measurements
should cover the high, intermediate and low RH ranges under humidification and dehumidification
conditions.

***Data availability.*** The measured results underlying this publication are available as online supplement.

*Competing interests. The authors declare that they have no conflict of interest.*

***Acknowledgements.*** This project was supported by the Strategic Priority Research Program (B) of the
Chinese Academy of Sciences (Grant No. XDB05010400), the National Key Research and Development
Program of China (2016YFC0202202), and the National Natural Science Foundation of China (Contract
No. 91544227, 41227805). The authors would like to thank J. Hong and Z. B. Wang for comments and
suggestions for improvement of the manuscript.

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

**Table 1.** Substances and their physical properties used in this work.

| Chemical compound | Chemical formula | Molar Mass [g mol⁻¹] | Density in solid or liquid state [g cm⁻³] | Solubility g/100cm³ H₂O | Solution surface tension [J m⁻²] | Manufacturer |
|---|---|---|---|---|---|---|
| Ammonium sulfate | $(NH_4)_2SO_4$ | 132.140 | 1.770[a] (solid), 1.550[a](liquid) | 74.400(at 20°C) | 0.072(0.001-10mg/mL) | Alfa Aesar, 99.95% |
| Levoglucosan | $C_6H_{10}O_5$ | 126.100 | 1.618[b](solid) 1.512[b](liquid) | | 0.073[e] (0.01-10mg/mL) | Aldrich, 99% |
| 4-Hydroxybenzoic acid | $C_7H_6O_3$ | 138.100 | 1.460(solid) 1.372[f](liquid) | 0.675(at 25°C) | 0.070[g] (>10mg/mL) | Alfa Aesar, 99.99% |
| Humic acid | | NA | 0.800[h](solid) | NA | NA | Aldrich, 99% |

[a]Clegg and Wexler, (2011);
[b]Lienhard et al, (2012);
[e]Tuckermann and Cammenga (2004) at 293K;
[f]Jedelsky et al, (2000);
[g]Kiss et al, (2005);
[h]Yates III and Wandruszka, (1999);











**Table 2.** The chemical composition of biomass-burning model mixtures studied given as mass
percentages (wt %).

| Mixture name | Levoglucosan | 4-Hydroxybenzoic acid | Humic acid | Ammonium sulfate |
|---|---|---|---|---|
| Mix-bio-dry | 87.2% | 9.2% | 1.5% | 2.1% |
| Mix-bio-wet | 68.0% | 26.0% | 3.0% | 3.0% |



















**Table 3.** Coefficients ($c_1$, $c_2$, $c_3$) of the fitted growth factor parameterization (Eq. 5) as follows:

| Chemical compounds | $c_1$ | $c_2$ | $c_3$ |
|---|---|---|---|
| Levoglucosan | 0.12868746 | 0.36582023 | -0.39840382 |
| 4-Hydroxybenzoic acid | -1.389967E-01 | 2.325586E-01 | -9.891943E-02 |
| Humic acid | -1.618304E-02 | 2.202483E-01 | 2.005134E-02 |


















**Table 4:** Experimental studies of organic and ammonium sulfate (AS) deliquescence and efflorescence
RH from this work and previous studies at 298K.

| Signal compound/Mixture | Organic mass fraction (%) | Deliquescence relative humidity of AS or organic in the mixed particle | Efflorescence relative humidity of AS or organic in the mixed particle |
|---|---|---|---|
| Levoglucosan | - | 80%[a]* | < 4%[a]* |
| | | 82.8%[b] | |
| | 25 | 80% | 45% |
| Levoglucosan+AS | 50 | - | - |
| | 75 | - | - |
| 4-hygroxybenzoic acid | - | > 97%[a]* | < 4%[a]* |
| | 25 | 80% | 35% |
| 4-Hydroxybenzoic acid+AS | 50 | 80% | 25% |
| | 75 | 80% | - |
| Humic acid | - | - | - |
| | 20 | 80% | 35% |
| Humic acid+AS | 50 | 80% | 35% |
| | 75 | 80% | 35% |

*is the DRH and ERH of pure organic components.
[a]Mochida and Kawamura. (2004)
[b]Zamora et al. (2011)





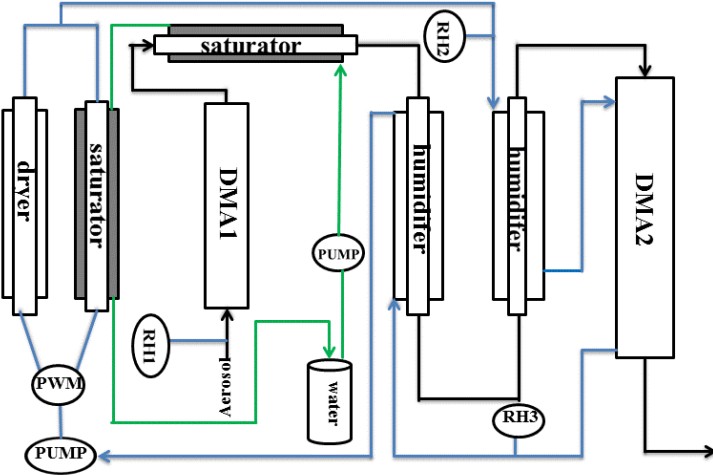


**Figure 1.** Schematic of the hygroscopicity tandem different mobility analyzer (HTDMA) system. the
sheath flow, aerosol flow, and water flow have been represented by the blue, black, green line,
respectively. PWM: Pulse Width Modulator circuit.






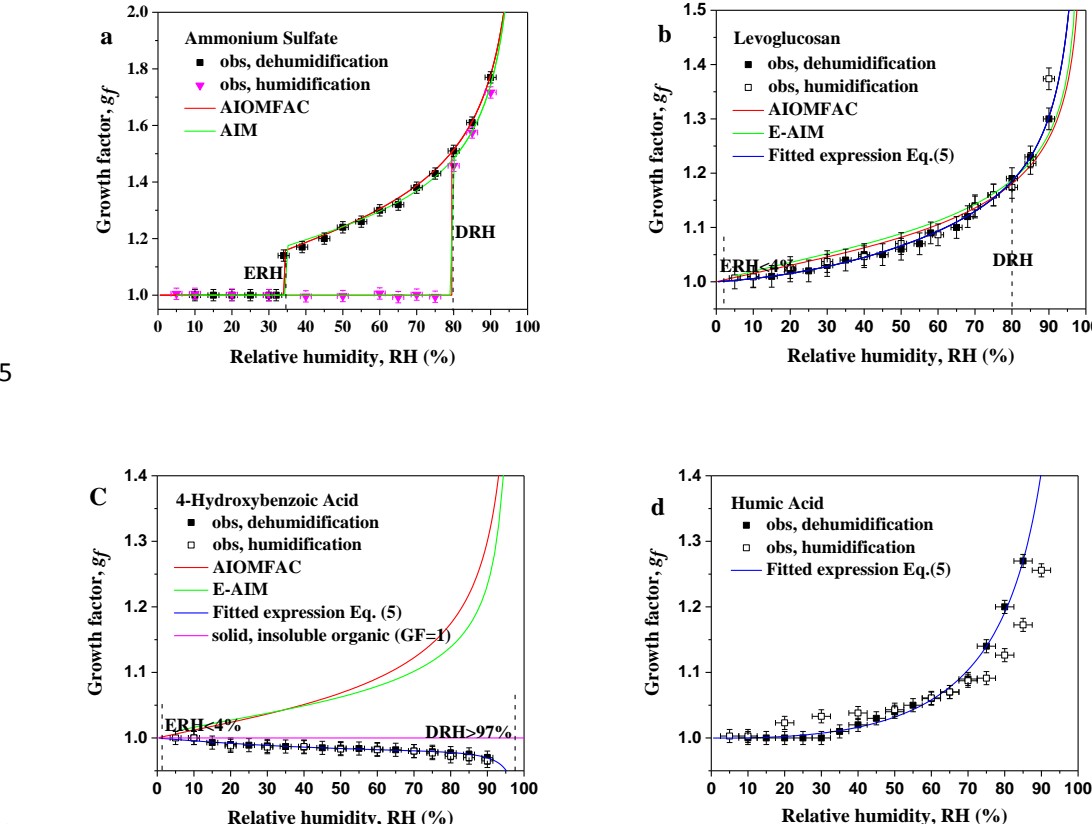



**Figure 2.** Hygroscopic growth, deliquescence and efflorescence of aerosol particles. Hygroscopic growth factors of (**a**) ammonium sulfate (AS), (**b**) levoglucosan, (**c**) 4-hydroxybenzoic acid, and (**d**) humic acid aerosol particles with dry diameter of 100 nm (open, black square). In this study, the green curves show E-AIM predictions, and the red curves the AIOMFAC predictions, and the blue lines the fitted expression (Eq. 5).





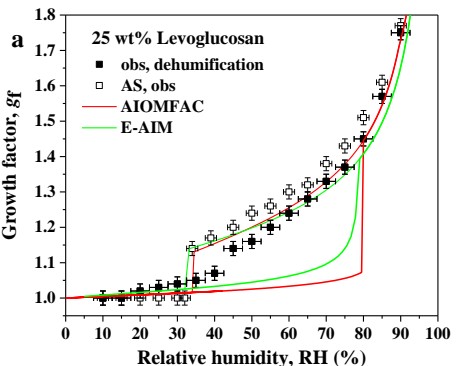
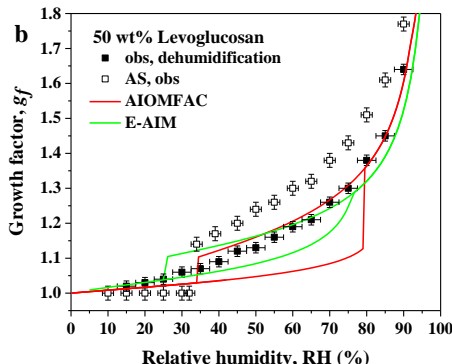


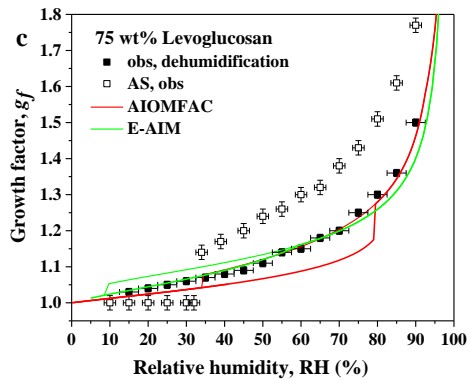

**Figure 3.** Hygroscopic growth, efflorescence of aerosol particles, and model predictions represent the
diameter growth factor during dehydration experiments in the range from 90 % to 5 % RH at 298.15 K.
(**a,b,c**). Hygroscopic growth curves of mixtures consisting of levoglucosan and ammonium sulfate (solid
symbols) at three different dry state mass fraction for particles of an initial dry diameter of 100 nm at RH
< 5 %) as compared to that of pure ammonium sulfate (open symbols, "AS, obs"). AIOMFAC-based
model predictions for bulk systems are shown in red, E-AIM predictions are shown in green.


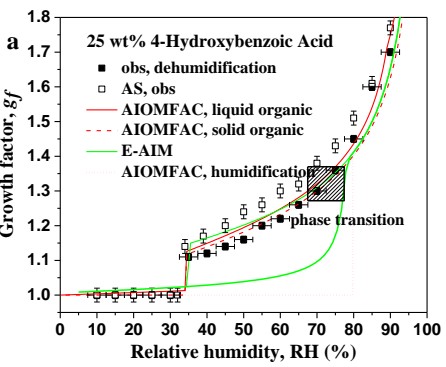

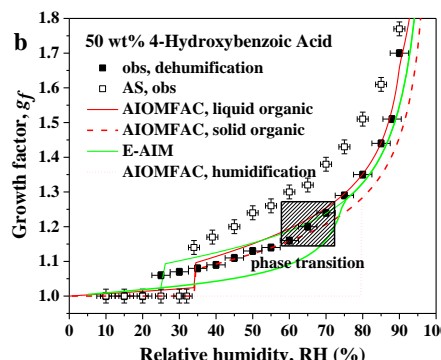


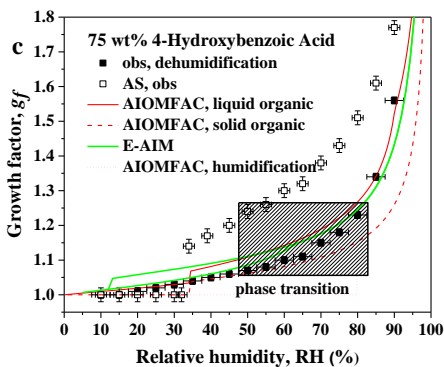


**Figure 4.** Hygroscopic growth factors, efflorescence of behavior, and model predications for dehydration
experiments in the range from 90 % to 5 % RH at 298.15 K. (**a,b,c**) hygroscopic growth curves of
mixtures consisting of 4-hydroxybenzoic acid and ammonium sulfate (solid symbols) at three different
dry state mass fraction (initial dry diameter of 100 nm at RH < 5 % ) as compared to that of pure
ammonium sulfate (open symbols). AIOMFAC-based model predictions for bulk systems are shown in
red, E-AIM-predictions are shown in green for the case of assuming that 4-hydroxybenzoic acid remains
in the liquid state. Shaded rectangle: RH range of gradual crystallization of 4-hydroxybenzoic acid.

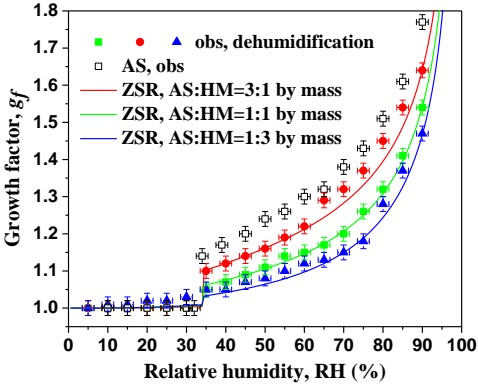


**Figure 5**. Hygroscopic growth factors, efflorescence of aerosol particles/constituents consisting of humic acid and ammonium sulfate at three different dry state mass fractions with initial dry diameter of 100 nm at RH < 5 % as compared to that of pure ammonium sulfate (open symbols). Colored curves: ZSR predictions of dimeter growth factors for dry particle compositions corresponding to the experimental data during dehumidification in the range from 90 % to 5 % RH at 298.15 K.

















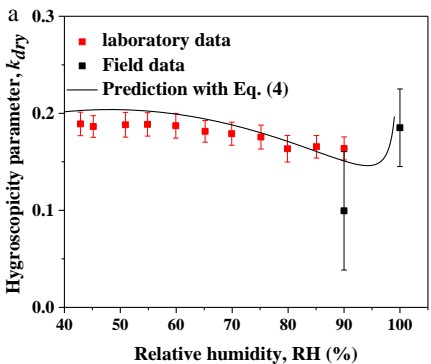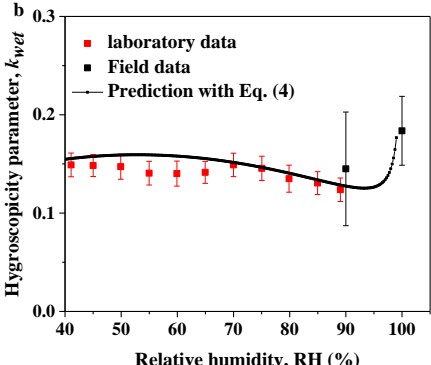


**Figure 6.** Hygroscopicity parameter, $\kappa$, representing mixed aerosol particles consisting of organic
surrogate components and ammonium sulfate at different periods (initial dry diameter of 100 nm at RH
< 5 %). The black curves in panels (a, b) show the $\kappa$ prediction from Eq. (4) with $HGF_{mix}$ calculated by
Eq. (6) using component volume fractions and the HGF of the individual mixture components from a fit
to the laboratory data (using Eq.5). the black symbols and error bars show field data from the Amazon
during the dry and wet periods at conditions of water vapor sub-saturation (HTDMA measurement) and
super-saturation ($\kappa_{CNN}$) (Whitehead et al., 2016; Pöhlker et al., 2006).

