# Peer review of "biomass burning and their effect on the efflorescence of"

_Atmospheric Chemistry and Physics, 2017_

## Referee Comment (RC1) · Anonymous Referee #1 · 19 Jul 2017

This paper discusses the measurements of hygroscopic growth of mixed biomass burning related organic proxys and AS using a HTDMA. The technique and data analysis approach are very standard. The highlight to me is the discussions of liquid phase change using AIOMFAC in interpreting the data and the relevance to BBOA. The scope of the paper suits ACP. My major concerns of the paper are on issues related to clarity and novelty.

1. Overall, the English is acceptable but some sentences, especially those in the abstract and the result discussions, are quite awkward. 2. A lot of these types of labora-

tory experiments have been carried out, dating back to when HTDMA was first developed by Peter McMurry. The authors should emphasize the novelty of the paper in the revision. The systems they chose have been somewhat studied by others in laboratory experiments. Comparison with the literature results are encouraged when appropriate. 3. Give justification (and references) to why 4-hydroxybenzoic acid was chosen as one of the chemicals studied. 4. Lines 29 and 30: It is an overstatement to say that the measurement-model agreement is due to composition-dependent consideration of crystallization of AS in the model prediction. Rather, it is only that such consideration can possibly explain the results. 5. Line 32-33 is awkward. Need to rewrite 6. Line 35-36: the authors claim that the consistency between measurements and predictions suggests " the similar O:C ratios and ammonium sulfate mass fraction in the laboratory and field observation conditions". This argument is misleading. The logic is wrong. 7. Has the HTDMA system been evaluated? More information of its performance is useful. 8. Line 330-333. While levoglucosan is deliquescent, the ERH of AS was found to shift to HIGHER RH when the mass fraction of levoglucosan increases. This appears to be a contradiction to the deliquescent nature of levoglucosan. Any explanation? 9. How are the current results compared to those of Chan et al. (2005) who also studied levoglucosan Chan and Chan (2006) who also reported the GF of humic like substances? 10. Line 341-343: The authors attributed the delaying or suppressing AS efflorescence to increased viscosity at moderate and low RH. I hope the authors can elaborate more on this. Conceptually delaying and suppressing can have slightly different meaning. Delaying implies a transport limitation while suppressing can imply a thermodynamic consideration. It seems that the authors mean delaying due to mass transfer effects of highly viscous droplets. By citing Zardini et al. (2008) is not adequate. More discussion of the conditions and time scale comparison would be useful. Chan and Chan (2006) examine the limitations of HTDMA system when there are mass transfer limitations in water uptake. 11. Line 377-379: These are only modeling predictions. The exact role of the acid on deliquescent properties of AS needs to be addressed better. The sentence needs to be rephrased. 12. Page 15: They authors

attempt to rationalize Figure 4 by two-liquid-to-solid phase transition. Subsequent discussion uses the term "liquid phase". What liquid phase do the authors refer? Organic or aqueous. I cautioned that without additional information, it is rather speculative to attempt to explain the behavior of the mixed particles, especially that the difference between the experimental data and either model predictions is not that large. 13. 3.3.1 Give more explanation on why the chosen mixtures would represent Amazon BBOA in dry and wet seasons. 14. Line 448-450. Cheung et al have examined GF of ambient atmospheric particles as a function of RH. They conclude that Kappa at high RH (like 90%) can be different from those at 50% or below. The authors can elaborate more on whether their observation is general for ambient measurements or just a characteristic of BBOA/AS. 15. The difference between Kappa based on subsaturated and supersaturated measurements can be due to sparingly soluble materials. See Huff Hartz et al (2006) and Chan et al. (2008) for more details. Does 4-hydroxybenzoic acid represent a sparingly soluble organic?

References: a. Chan and Chan (2005), EST, 37 (22), 5109-5115 b. KE Huff Hartz (2006), AE 40 (4), 605-617 c. Chan and Chan (2005), ACP, 5 (10), 2703-2712 d. Chan et al. (2005), EST, 39 (6), 1555-1562 e. Chan et al. (2008) EST, 42 (10), 3602-3608. f. Cheung et al. (2015) AST, 49 (8), 643-654.

---

## Referee Comment (RC2) · Anonymous Referee #2 · 1 Sep 2017

This study presents an attempt to distinguish the role of particle phase state on the hygroscopicity of biomass burning surrogates and mixtures with ammonium sulphate. I consider the addition of such studies to the literature worthwhile. However, this paper requires a number of changes and clarifications before being accepted for publication. Before these are clarified, I found it difficult to provide further critique on a number of results presented. After reading the first review, which I tend to agree with on specific points raised, I present a number of different factors the authors need to address below:

Specific comments:

[Figure]

Abstract: I would recommend removing reliance on the word 'slightly'. Please quantify 'slightly' or remove entirely. This paper often feels a little too qualitative in nature, and I would recommend checking all instances like this. The authors also comment on 'similarity of hygroscopic parameter k'. Please quantify this. What range do you consider to be similar? line 35: Presume the authors mean sub-saturated 'RH'. line 35: 'at' the same environment? This is unclear. I would recommend checking all grammar throughout the document, as also noted by the other referee.

Page 7, line 176: 'Here the AIOMFAC-based thermodynamic equilibrium model is used to calculate the DRH...in the multicomponent system based on the known solubility of AS in the organic-free system.? I'm not sure why you have chosen to do this when the benefit of the AIOMFAC activity coefficient model is to account for inorganic-organic interactions. Please justify this as the proceeding equations do not necessarily correlate with this statement.

Page 8: line 209 'Differences in the density models are expected to lead to relatively small differences..' This needs a qualifying reference or a demonstration of sensitivity. What do you mean by relatively small?

Page 10, line 272: 'standard UNIFAC..'. Which set of interaction parameters are you using? A reader should be able to replicate these results.

Page 10, line 276:'intra molecular interactions are fully considered by these models..'. What is UNIFAC based on?

Page 18, line 482:'at RH > 95% the water content of hygroscopic particles increases dramatically with a small increase in RH, leading to the predicted change in the mixtures k parameter that is best representing the hygroscopic growth under such high-RH conditions'. This statement is not clear. Are you suggesting that variable 'k' values are required? Please rephrase and clarify.

General comments:

Please add a brief discussion on the expected performance of activity models for 4-hydroxybenzoic acid. It would help the reader understand where sensitivities might lie.

Have the authors considered how a variable morphology would influence results? Is there no literature data on studies using AIOMFAC on this?

What is the residence time of particles in the HTDMA? If there were a phase state change from which kinetic mass transfer limitations might apply, how might this change your conclusions?

I would appreciate more discussion on how the reliance on 3 organic surrogates influences conclusions for a SOA class that is likely much more complex. Are you studies sensitive to complexity, influenced by a discrete range of solubilities and 'step-like' transitions? How would you test this?

The authors need to follow Copernicus guidelines on accessibility of data and software before this paper is accepted for publication. Please read:https://www.atmospheric-chemistry-and-physics.net/about/data_policy.html

---

## Author Comment (AC1) · 27 Oct 2017

Dear referee,

Thank for your comments. please find the attached about response to comments from referee #1, and revised manuscript.

thank you very much.

Ting

[Figure]

Please also note the supplement to this comment:
https://www.atmos-chem-phys-discuss.net/acp-2017-549/acp-2017-549-AC1-supplement.zip

———————————————————

---

## Author Comment (AC3) · 1 Nov 2017

Dear referees

Thank you for your comments for our manuscript.

please find the attached about my responses to referee #1, #2, and revised manuscript in the supplement.

have a nice day

[Figure]

Ting

Please also note the supplement to this comment:
https://www.atmos-chem-phys-discuss.net/acp-2017-549/acp-2017-549-AC3-
supplement.zip

———————————————————

---

## Editor Comment (EC1) · M. D. Petters (Editor) · 11 Nov 2017

This comment is in response to a message sent by one of the referees. The referee made a mistake in one of his/her comment (see below). I encourage the authors to respond to this correction while the manuscript is undergoing a second round of reviews.

Original referee comment

8. Line 330-333. While levoglucosan is deliquescent, the ERH of AS was found to shift to HIGHER RH when the mass fraction of levoglucosan increases. This appears to be

a contradiction to the deliquescent nature of levoglucosan. Any explanation?

This was meant to say NON-deliquescent. The intended question was

8. Line 330-333. While levoglucosan is non deliquescent, the ERH of AS was found to shift to HIGHER RH when the mass fraction of levoglucosan increases. This appears to be a contradiction to the Non deliquescent nature of levoglucosan. Any explanation?

---

## Author Response (AR1)

**Authors' response to comments by anonymous referee #1:**

*This paper discusses the measurements of hygroscopic growth of mixed biomass burning related organic proxys and AS using a HTDMA. The technique and data analysis approach are very standard. The highlight to me is the discussions of liquid phase change using AIOMFAC in interpreting the data and the relevance to BBOA. The scope of the paper suits ACP. My major concerns of the paper are on issues related to clarity and novelty.*

**Response:** We would like to thank referee #1 for the comments and the constructive suggestions, which help us to improve our manuscript. We address in the following the comments and suggestions by referee #1 and provide improvements based on these clarify the questioned issues in the revised manuscript.

In this study, we focus on two distinct aspects of hygroscopicity-related research: (1) well-defined hygroscopic diameter growth factor measurements and associated modeling for simple organic-inorganic mixtures; (2) the comparison of the effective hygroscopicity parameter ($\kappa$) of relatively simple mixtures of representative organic biomass burning surrogates to hygroscopicity values obtained from field data.

The experimental observations with well-defined organic-inorganic laboratory systems suggests that such systems exhibit considerable variability with regard to solid-liquid phase transitions during dehumidification-an aspect that is rarely focused on in previous published work. For example, in the case of the 4-hydroxybenzoic acid + ammonium sulfate system, the data interpretation from our measurement-model comparison suggests the occurrence of two-solid-liquid phase transitions at different RH level, one involving a solid organic phase and the other the crystallization of ammonium sulfate. The comparison presented for the effective hygroscopicity parameter of our surrogate mixtures for biomass burning particles, e.g. $\kappa \approx 0.12$ to $0.15$ for the wet-season mixture in the 90 % to 40 % RH range, shows good agreement with field data for the wet season in Amazon (WSOC $\kappa \approx 0.14 \pm 0.06$ at 90 % RH). This suggests that laboratory-generated mixtures containing organic surrogate compounds and ammonium sulfate can be used to mimic, in a simplified manner, the chemical composition of ambient aerosols from the Amazon for the purpose of RH-dependent hygroscopicity studies.

*1. Overall, the English is acceptable but some sentences, especially those in the abstract and the result discussions, are quite awkward.*

**Response**: Many thanks, we have carefully revised the manuscript regarding language issues, including grammar, wording, and sentence structure. For example, following the reviewer's suggestions, we have made changes in the following places and they now reads as:

**Page 1 line 17:** "We observed that levoglucosan and humic acid aerosol particles release water upon dehumidification in the range from $90-5$ % relative humidity (RH)."

**Page 1 line 18:** "4-Hydroxybenzoic acid aerosol particles, however, remain in the solid state both upon humidification or dehumidification and exhibit a small shrinking in size at higher RH compared to the dry size. For example, the measured growth factor of 4-hydroxybenzoic acid aerosol particles is ~0.96 at 90 % RH."

**Page 2 line 32-37: "**A similarity of hygroscopic parameter $\kappa$ for organic surrogate compounds mixed with ammonium sulfate for different mass fractions during the different seasonal periods in the Amazon is observed………,which suggests the similar O: C ratios and ammonium sulfate mass fraction in the laboratory and field observation conditions." **was revised to**:

"Lastly, two distinct mixtures of organic surrogate compounds, including levoglucosan, 4-hydroxybenzoic acid, and humic acid were used to represent the average water-soluble organic carbon (WSOC) fractions observed during the wet and dry seasons in the central Amazon Basin. A comparison of the organic fraction's hygroscopicity parameter for the simple mixtures, e.g. $\kappa \approx$ 0.12 to 0.15 for the wet-season mixture in the 90 % to 40 % RH range, shows good agreement with field data for the wet season in Amazon (WSOC $\kappa \approx 0.14\pm0.06$ at 90 % RH). This suggests that laboratory-generated mixtures containing organic surrogate compounds and ammonium sulfate can be used to mimic, in a simplified manner, the chemical composition of ambient aerosols from the Amazon for the purpose of RH-dependent hygroscopicity studies."

**Page 5 line 124-126:** "Three organic compounds, levoglucosan, 4-hydroxybenzoic acid and humic acid, were chosen to systematically study the influence…..and mixed organic-AS-containing aerosol particles" **was revised to:**

"The three organic compounds levoglucosan, 4-hydroxybenozic acid and humic acid were used as surrogates for the rich-class of water-soluble organic components in biomass burning aerosols. The influence of the distinct chemical structure of these compounds was studied with regard to the water uptake and evaporation behavior of the pure organic compounds as well as for mixed organic-AS-containing particles. Furthermore, a comparison with field data from the Amazon was preformed to quantify the ability of mixtures of these three organic compounds in mimicking the hygroscopic properties of complex ambient organic particles originating from biomass burning emissions."

**Page 10 line 284-287, we revise:** "Also, no ERH of 4-hydroxybenzoic acid in the dehydration mode was observed during the experiments; the likely reason is that the highest RH reached in the humidifier was approximately 98 %, which may be below the ERH of 4-hydroxybenzoic acid, reported as above 95 % RH in another study (Mochida and Kawamura, 2004)."

**Page 13 line 345, we revise:** "The E-AIM prediction is in relatively good agreement with results from the HTDMA measurement but typically overestimates the water content of particles consisting of organic-AS mixtures at the RH range close to the ERH of AS."

*2. A lot of these types of laboratory experiments have been carried out, dating back to when HTDMA was first developed by Peter McMurry. The authors should emphasize the novelty of the paper in the revision. The systems they chose have been somewhat studied by others in laboratory experiments. Comparison with the literature results are encouraged when appropriate.*

**Response**: with respect to a comparison of our measured results with published data, we have added some discussion to the revised manuscript to highlight the difference/agreement between our measurements and previous results. For example, levoglucosan amount to a substantial portion of the organic aerosol mass and its hygroscopic behavior has been studied by several groups in the past (e.g., Mochida and Kawamura, 2004; Chan et al., 2005; Svenningsson et al., 2006). However, the effect of levoglucosan on the hygroscopic behavior of AS-containing aerosol particles as well as crystallization of AS in the mixed system have been studied rarely by previous papers (Maskey et al., 2014). Most studies concerning the hygroscopic behavior of HULIS have employed certain model compounds, mainly terrestrial and aquatic humic and fulvic acids (Brooks et al., 2004; Chan and Chan, 2003; Sjogren et al., 2007). All of these studies suggest that the measured variable

hygroscopicities of the HULIS can be attributed to the different sources (molecular structure) of the selected compounds. Mono- and dicarboxylic acids account for a significant mass fraction of the organic mass portion of biomass burning particles (e.g., Hoffer et al., 2006; Dusek et al., 2011), their pure-component hygroscopic behavior or effects on the hygroscopic behavior of mixed organic-inorganic particles have been measured by several groups (Chan et al., 2005; Hanford et al., 2008; Zandin et al., 2008). We summarize that these acids have significant effects on the deliquescence of ammonium sulfate. Also, there are rarely corresponding model predictions available for the DRH and ERH of AS in organic + AS mixed particles. in our study, such model predictions were carried out to further discuss model-measurement agreement/deviations and the interpretation of the experimental data based on theory. Table 1 (shown below) summarizes deliquescence and efflorescence properties of various organic compounds and mixtures with ammonium sulfate of relevance for a comparison of these phase transitions with the systems used in this study.

**Table 1. Deliquescence relative humidity and efflorescence relative humidity of pure organic compounds or of AS in mixtures reported by difference references. Most data taken at a temperature in the range from 290-300K.**

| Single compound/mixture | DRH of organic* or AS in the mixed particle (%) | ERH of organic* or AS in the mixed particle (%) | Reference |
|---|---|---|---|
| Levoglucosan | 80* | < 4 * | Mochida and |
| D-glucose | 90* | < 4 * | Kawamura (2004) |
| Syringic acid | > 97* | - | |
| Vanillic acid | > 97* | - | |
| 4-Hydroxybenzoic acid | > 97* | < 4 * | |
| L-glycine | 93.2* | 53.6 - 55.2 * | Chan et al. (2005) |
| L-alanine | 96.9* | 67.3 - 70.8 * | |
| L-serine | 99.1* | 27.6 - 30.8 * | |
| L-glutamine | 98.8* | 72.1 - 74.7 * | |
| L-arginine | - | 42.2 - 47.3 * | |
| L-asparagine | 99.5* | - | |
| Levoglucosan | - | - | |
| Mannosan | - | - | |
| Galactosan | - | - | |

| | | | |
|---|---|---|---|
| Malonic acid | No | No | Peng et al. |
| Oxalic acid | No | 51.8 - 56.7* | (2001) |
| Succinic acid | No | 55.2 - 65.7* | |
| Glutaric acid | 83.5 - 85* | 29 - 33* | |
| Citric acid | No | No | |
| DL-malic acid | No | No | |
| L-(+)-tartaric acid | No | No | |
| D-Glucose | 88.1±1.8* | - | Zamora et al. |
| Levoglucosan | 82.8±1.8* | - | (2011) |
| Succinic acid | 99.5±1.8* | - | |
| Phthalic acid | > 99.5* | - | |
| Glutaric acid | 90.1±1.8* | - | |
| Oxalic acid | 97.9±1.8* | - | |
| Suwannee River fulvic acid | > 99.9* | - | |
| Nordic Reference fulvic acid | > 99.9* | - | |
| Humic acid | > 99.9* | - | |
| Nordic Aquatic Fulvic Acid (NAFA) | No | No | Chan and Chan |
| NAFA + AS (mass ratio of 1:1) | 78.6 - 81.4 | 38.1 - 41.5 | (2003) |
| Suwannee River Fulvic Acid (SRFA) | No | No | |
| SRFA + AS (mass ratio of 1:1) | 79.8 - 82.5 | 39.8 - 43.6 | |
| Fluka chemical | 70* | - | Brooks et al. |
| Fluka + AS (10 wt% Fluka) | 78 | - | (2004) |
| Fluka + AS (53 wt% Fluka) | 69 | - | |
| Pahokee Peat Reference | 50* | - | |
| Pahokee Peat Reference + AS (2wt% HA) | 79 | - | |
| Pahokee Peat Reference + AS (13wt% HA) | 79 | - | |
| Leonardite Standard | 70* | - | |
| Leonardite Standard + AS (2wt% HA) | 70 - 80 | - | |
| Leonardite Standard + AS (13wt% HA) | 70 - 80 | - | |
| Suwannee River Reference | No | - | |
| Suwannee River Reference + AS(8wt%FA) | 70 - 80 | - | |
| Suwannee River Reference + AS(wt%FA) | 70 - 80 | - | |
| Citric Acid | No | No | Zardini et al. |
| Citric Acid + AS(molar ratio of 1:4) | 78 | 37 | (2008) |
| Glutaric Acid | 90* | 43* | |
| Glutaric Acid + AS (molar ratio of 1:1) | 77 | 32 | |
| Adipic Acid | No | No | |
| Adipic Acid + AS (molar ratio of 3:1) | 78 - 83 | 33 - 69 | |

| | | | |
|---|---|---|---|
| Adipic Acid | No | No | Sjogren et al. |
| Adipic Acid + AS (mass ratio of 1:1) | 80 | 32 | (2006) |
| Humic acid | No | No | |
| Humic acid + AS (mass ratio of 1:3) | No | No | |
| Glycerol+AS (mass ratio of 1:1) | 77.8 | 28.2 | Choi and Chan |
| Succinic acid + AS (mass ratio of 1:1) | 79.0 | 48.3 | (2002) |
| Malonic acid + AS (mass ratio of 1:1) | 73.7 | 45.9 - 46.9 | |
| Citric acid+AS (mass ratio of 1:1) | No | No | |
| Glutaric Acid + AS (mass ratio of 1:1) | 76.6 | 55.7 - 59.2 | |
| AS particles coated with 49 wt% glutaric acid | 80.9[1] 79[2] | 40.9[1] 40.6[2] | Chan et al. |
| AS particles coated with 13 wt% glutaric acid | 80.7[1] 81.1[2] | 50.3[1] 51.1[2] | (2006) |
| Levoglucosan + AS (mass ratio of 1:1) | No | No | Lei et al. |
| Humic acid + AS (mass ratio of 1:1) | 80 | 25 | (2014,) this |
| 4-Hydroxybenzoic acid + AS (mass ratio of 1:1) | 80 | 35 | study |

*is the DRH and ERH of pure organic components

[1]is the first humidity cycle

[2]is the second humidity cycle

No: no observation in the study

-: no study

**Related additions and changes included in the revised manuscript:**

**Page 5 line 124-126:** "Three organic compounds, levoglucosan, 4-hydroxybenzoic acid and humic acid, were chosen to systematically study the influence…..and mixed organic-AS-containing aerosol particles" **was revised to:**

"The three organic compounds levoglucosan, 4-hydroxybenozic acid and humic acid were used as surrogates for the rich-class of water-soluble organic components in biomass burning aerosols. The influence of the distinct chemical structure of these compounds was studied with regard to the water uptake and evaporation of the pure organic compounds as well as for mixed organic-AS-containing particles. Furthermore, a comparison with field data from the Amazon was preformed to quantify the ability of mixtures of these three organic compounds to mimicking the hygroscopic behavior of complex ambient organic particles originating from biomass burning emissions."

**Page 10 Line 264-268: we add the reference Chan et al. (2005) and modify the text slightly.**
"The similarity of diameter growth factors both under hydration and dehydration conditions even below the DRH of levoglucosan is explained by the lack of crystallization of levoglucosan upon

drying to low RH and the presence of a metastable supersaturated aqueous levoglucosan solution in both the hydration and dehydration modes for experiments initiated with liquid solution droplets (Mochida and Kawamura, 2004; Chan et al., 2005; Svenningsson et al., 2006)."

**Page 10 Line 269-271: we add Chan et al. (2005).** "A possible reason for a persistent metastable supersaturated solution states is that levoglucosan particles remain liquid (possibly a viscous liquid state) upon drying to below 5 % RH, which was also observed previously with a reported ERH < 4% RH (Mochida and Kawamura, 2004; Chan et al., 2005)."

**Page 11 line 305-306: we add the reference Chan and Chan. (2005) and modify the text slightly**. "For example, a stepwise change in the water absorption and desorption behavior within different RH range was observed in the case of Nordic Aquatic Fulvic Acid (NAFA) and Suwannee River Fulvic Acid (SRFA) by Chan and Chan (2005). These hygroscopic behaviors suggest that humic acid particles and structurally similar compounds remain some water down to the low RH levels achieved in the instruments (imperfect drying during particle residence in the instrument)."

*3. Give justification (and references) to why 4-hydroxybenzoic acid was chosen as one of the chemicals studied.*

**Response**: Mochida and Kawamura (2004) have previously studied some surrogates of the organic fraction observed in biomass burning aerosols based on field data. Among many organic acids, 4-hydroxbenzoic acid was detected in fine model aerosols in the Amazon region during field campaigns (Decesari et al., 2006; Rissler et al., 2006). They observed that a significant fraction of WSOC are aromatic acids and 4-hydroxybenzoic acid can be considered a surrogate for the wide class of similar aromatic acids in such aerosols. The physicochemical properties, e.g. water-solubility and volatility have been extensively studied previously (Mochida and Kawamura, 2004; Hoffer et al., 2006; Dusek et al., 2011; Lei et al., 2014). However, its effect on the hygroscopic behavior of organic or organic-inorganic mixtures, as well as its impact on the DRH and ERH of inorganics such as ammonium sulfate and sodium chloride have not yet been reported. Therefore, in our study, 4-hydroxybenzoic acid is selected to represent a group of lignin pyrolysis products.

*4. Lines 29 and 30: It is an overstatement to say that the measurement-model agreement is due to composition-dependent consideration of crystallization of AS in the model prediction. Rather, it is only that such consideration can possibly explain the results.*

**Response**: Thanks and we agree with the reviewer and have modified the text:

**Page 1-2 line 28-30:** "In addition, consideration of a composition-dependent degree of dissolution of crystallization AS (solid-liquid equilibrium) in the AIOMFAC and E-AIM models leads to a relatively good agreement between models and observed growth factors, as well as ERH of AS in the mixed system."

*5. Line 32-33 is awkward. Need to rewrite.*

**Response: Page 2 line 32-37, "**A similarity of hygroscopic parameter $\kappa$ for organic surrogate compounds mixed with ammonium sulfate for different mass fractions during the different seasonal periods in the Amazon is observed………,which suggests the similar O: C ratios and ammonium sulfate mass fraction in the laboratory and field observation conditions."

**We rewrote it as:** "Lastly, two distinct mixtures of organic surrogate compounds, including levoglucosan, 4-hydroxybenzoic acid, and humic acid were used to represent the average water-soluble organic carbon (WSOC) fractions observed during the wet and dry seasons in central Amazon Basin. A comparison of the organic hygroscopicity parameter for the simple mixtures, e.g. $\kappa \approx 0.12$ to $0.15$ for the wet-season mixture in the 90 % to 40 % RH range, shows good agreement with field data for the wet season in Amazon (WSOC $\kappa \approx 0.14\pm0.06$ at 90 % RH). This suggests that laboratory-generated mixtures containing organic surrogate compounds and ammonium sulfate can be used to mimic, in a simplified manner, the chemical composition of ambient aerosols from the Amazon for the purpose of RH-dependent hygroscopicity studies."

*6. Line 35-36: the authors claim that the consistency between measurements and predictions suggests " the similar O:C ratios and ammonium sulfate mass fraction in the laboratory and field observation conditions". This argument is misleading.*

**Response**: we have revised the sentence in the manuscript.

**Page 2 line 34-36, we rewrote it as:** "This suggests that laboratory-generated mixtures containing organic surrogate compounds and ammonium sulfate can be used to mimic, in a simplified manner, the chemical composition of ambient aerosols from the Amazon for the purpose of RH-dependent hygroscopicity studies."

*7. Has the HTDMA system been evaluated? More information of its performance is useful.*

**Response**: The performance of employed HTDMA system has been reported in previous studies (Lei et al., 2014, Jing et al., 2015, 2017; Liu et al., 2015; Peng et al., 2016). The aerosol particle sizing by the DMA as well as the RH measurement were calibrated independently. Sizing is crucially dependent on correct flow rate and high voltage (HV). All flows and analogue output for controlling the HV supply as well as the HV amplification factor were calibrated in the system. The performance of the DMA can be tested with certified particles of known size such as polystyrene latex (PSL) spheres (e.g., 203, 145, 100, 60 nm PSL). All RH sensors in the HTDMA system were calibrated with a reference dew point sensor. Also, validation of the HTDMA was carried out by using standard substances (e.g., ammonium sulfate, sodium chloride) to test the accuracy of the system in the RH range from 5 - 90 %. For example, the calibration results for two RH sensors and for the size determination by the DMA is shown in the following figures:

[Figure]

**Fig.1** RH sensors calibration with a reference dew point hygrometer

[Figure]

**Fig.2** Size calibration of DMA₂ measurement with polystyrene latex (PSL) spheres of known diameter

[Figure]

**Fig.3** Test of the HTDMA accuracy with ammonium sulfate at 298.15K

**Related additions and changes included in the revised manuscript:**

**Page 5 line 135, we add:** "Figure 1 shows a schematic of the HTDMA instrument, more detailed information about this instrument's setup, calibration and evaluation is described elsewhere (Lei et al., 2014; Jing et al., 2015, 2017; Liu et al., 2015). Briefly,"

*8. Line 330-333. While levoglucosan is deliquescent, the ERH of AS was found to shift to HIGHER RH when the mass fraction of levoglucosan increases. This appears to be a contradiction to the deliquescent nature of levoglucosan. Any explanation?*

**Response**: We do not agree with the referee's comment about an apparent contradiction. The statement in the text is about the ERH of AS (not the DRH). The observation that the presence of levoglucosan affects this ERH, in one case by causing crystallization of AS at a higher RH than in the pure AS case, is not a contradiction. Since the liquid solution with all of AS dissolved exists as a metastable state below the DRH of AS during dehumidification, in principle, crystallization could occur at any RH below the DRH in complete agreement with thermodynamic theory. Crystallization of AS from the supersaturated solution (below its DRH for the given mixture) is essentially a stochastic process. It is more likely to occur towards higher supersaturation (lower RH), but is possible also at higher RH than in the case of homogeneous nucleation of AS crystals in pure AS particles. Levoglucosan may inhibit crystallization (as seen for the mixtures with higher levoglucosan mass fraction) or it may promote it, which may depend on its mass fraction in the solution, which impacts the mixtures viscosity as well as the ion activity product of AS at any particular RH.

**Related additions and changes included in the revised manuscript:**

**Page 12 line 330-333: we clarify:** "When the concentration of levoglucosan is low (25 wt %), a clear efflorescence of AS can be found but shift to a higher RH (40 % - 45 %) than the ERH of pure AS (Fig.3a). Similar phenomenon has been also found for the mixed system of NaCl and Nordic Aquatic Fulvic Acid (NAFA) that the crystallization of NaCl shifted to higher RH by mixing with NAFA at a mass ratio of 1:1 (Chan and Chan. 2003). With increasing mass fraction of levoglucosan (i.e., 50 wt % and 75 wt %), the mixtures release water gradually, no crystallization of AS was observed. Although a small step of crystallization of AS may still happen, it can not be detected by our system."

9. How are the current results compared to those of Chan et al. (2005) who also studied levoglucosan Chan and Chan (2006) who also reported the GF of humic like substances?

**Response:** Thanks for pointing to these papers. Chan et al. (2005) studied the hygroscopic behavior of pure levoglucosan using an electrodynamic balance (EDB). They observed continuous water absorption and desorption in the range from 0 to 100 % RH, indicating no phase transition of levoglucosan occurring during the humidification and dehumidification process in the EDB. We also compared HGFs obtained by our HTDMA with the HGFs reported by other groups (Mochida and Kawmura., 2004; Chan et al., 2005) shown in Figure 4 below. They agree well with in the RH range we studied (RH < 90 %). For example, the measured diameter growth factors of levoglucosan particles at 80 % RH are 1.18 using our HTDMA, the same value measured by Mochida and Kawmura (2004) with their HTDMA, and 1.15 measured by the EDB technique of Chan et al. (2005).

Chan and Chan (2003) studied the hygroscopicity of humic acid and its mixture with NaCl and AS. We note that the hygroscopicity of humic acid is sensitive to the type of humic acid studied (actual molecular strcture). We show that the measured diameter growth factor of humic acid lies between the observed growth factors of two distinct types of humic acids determined by Chan and Chan (2003) (Fig. 4. right).

 **Related additions and changes included in the revised manuscript:**

**Page 10 Line 264-268: we add the reference Chan et al. (2005) and modify the text slightly.**

"The similarity of diameter growth factors both under hydration and dehydration conditions even below the DRH of levoglucosan is explained by the lack of crystallization of levoglucosan upon drying to low RH and the presence of a metastable supersaturated aqueous levoglucosan solution

in both the hydration and dehydration modes for experiments initiated with liquid solution droplets (Mochida and Kawamura, 2004; Chan et al., 2005; Svenningsson et al., 2006)."

**Page 10 Line 269-271: we add Chan et al. (2005).** "A possible reason for a persistent metastable supersaturated solution states is that levoglucosan particles remain liquid (possibly a viscous liquid state) upon drying to below 5 % RH, which was also observed previously with a reported ERH < 4% RH (Mochida and Kawamura, 2004; Chan et al., 2005)."

**Page 11 line 305-306: we add the reference Chan and Chan. (2005) and modify the text slightly.** "For example, a stepwise change in the water absorption and desorption behavior within different RH range was observed in the case of Nordic Aquatic Fulvic Acid (NAFA) and Suwannee River Fulvic Acid (SRFA) by Chan and Chan (2005). These hygroscopic behaviors suggest that humic acid particles and structurally similar compounds remain some water down to the low RH levels achieved in the instruments (imperfect drying during particle residence in the instrument)."

[Figure]

**Fig 4**. Comparison of our measurements and previous observations for the hygroscopic diameter growth factor of levoglucosan and humic acid aerosol particles.

*10. Line 341-343: The authors attributed the delaying or suppressing AS efflorescence to increased viscosity at moderate and low RH. I hope the authors can elaborate more on this. Conceptually delaying and suppressing can have slightly different meaning. Delaying implies a transport limitation while suppressing can imply a thermodynamic consideration. It seems that the authors mean delaying due to mass transfer effects of highly viscous droplets. By citing Zardini et al. (2008) is not adequate. More discussion of the conditions and time scale comparison would be useful. Chan and Chan (2006) examine the limitations of HTDMA system when there are mass transfer limitations in water uptake.*

**Response:** We thank the referee for mentioning this distinction in terminology. We agree that "delaying" or impeding of efflorescence fits better in this context-albeit a completed "suppression", as stated by Zardini et al. (2008), cannot be ruled out as a possibility. Reference to Zardini et al. (2008) is adequate in our opinion since that article discusses such an observed effect for citric acid + AS mixtures (from the perspective of both HTDMA and EDB experiments). However, we add further discussion on the effect of potential mass transfer limitations for aqueous aerosol HGF measurements and for the solid-liquid phase transitions during deliquescence.

**Related additions and changes included in the revised manuscript:**

**Page 13 line 335-343, we add:** "the rather high viscosity of solutions containing levoglucosan is expected to increase considerably toward RH (Marshall et al., 2016). This increase in viscosity might impede the crystallization of AS in the mixed systems on the time scale of the experiment. Mass transfer limitation effects on the deliquescence or efflorescence process of (crystalline) organic particles and the water uptake or evaporation have been investigated in several experimental studies (Peng et al., 2001; Choi and Chan, 2002; Chan and Chan, 2005; Sjogren et al., 2007; Zardini et al., 2008; Ciobanu et al., 2010; Smith et al., 2012; Mikhailov et al., 2013; Hodas et al., 2015). Mass transfer limitations may impact the outcome of experiments significantly if the characteristic time scale for equilibration is similar to or larger than the residence time of particles in the experimental setup. In this study, the total residence time of the aerosol sample during the equilibration phase before entering the DMA2 is about 8 s. In order to improve the probability that the particles reach equilibrium with the target RH during this residence time, the monodisperse aerosol selected by DMA1 is first humidified to 98 % RH. The aerosol particles are then exposed to a lower target RH by a two-step process using double Nafion tubes. Kerminen (1997) estimated the necessary residence time for achievement of water equilibrium of aqueous droplets to be between $8 \times 10^{-6}$ s and 0.1 s for 100 nm and 500 nm particles, respectively. Therefore, the typical residence time of a few seconds in the humidification or dehumidification section in a HTDMA measurement is assumed to be sufficient for most equilibrium hygroscopicity measurements (Brooks et al., 2004; Mikhailov et al., 2004). Moreover, our HGF results for the three pure organic components are in good agreement with respective data by Mochida and Kawamura, (2004), Brooks et al., (2004) and Chan and Chan (2005), conducted with different techniques and/or residence times. However, there are cases where water equilibration could be impeded substantially in the presence of highly viscous or glassy particles at low RH, e.g. for

ternary sucrose + NaCl + water particles of > 6 μm in diameter studied by Bones et al. (2012), who report an equilibration time scale > 1000 s for such particles. Note that, aside from viscosity, there is an important size-dependence of the particles on the equilibration time scale (e.g. Koop et al. 2011). For aqueous 100 nm particles used in HTDMA experiments at room temperature, Bones et al. (2012) indicate that the equilibration time scale for water is likely only of concern for RH < 10 % in such an instrument. We therefore conclude that the residence time of 8 s is very likely sufficient to allow for equilibrium HGF measurements in dehydration mode, at least down to 10 % RH (when starting with aqueous solution droplets).

Mass transfer effects in hygroscopicity measurements of aerosol particles during hydration conditions have been encountered previously, particularly when a solid-liquid phase transition (deliquescence) is involved (Peng et al., 2001; Chan and Chan., 2005; Sjogren et al., 2006). For example, Peng et al. (2001) observed in electrodynamic balance (EDB) experiments under conditions of very slow humidification that glutaric acid aerosol particles showed a deliquescence phase transition in the RH range from 83 to 85 % over the course of several hours. This is a much longer time span than that of ~ 40 min for the deliquescence of other super-micron sized dicarboxylic acid particles (e.g., malonic acid) in EDB experiments. This observation indicates that the solid-liquid phase transition of glutaric acid particles may likely be mass transfer limited during the hydration process. In this context, it is possible that the deliquescence of initially solid, pure 4-hydroxybenzoic acid particles at RH > 97 % is further impeded by slow dissolution, which could have led to the absence of deliquesced particles (Fig. 2c) on experimental time scale."

*11. Line 377-379: These are only modeling predictions. The exact role of the acid on deliquescent properties of AS needs to be addressed better. The sentence needs to be rephrased.*

**Response**: The discussions about the role of the acid on the deliquescent properties of AS was actually based on HTDMA results of Lei et al. (2014) and shown in **Fig.6**. We have clarified this in the revised manuscript. Following the reviewer's suggestion, we add a bit more discussion on the role of the different types of acids on the deliquescent phase transition of AS in the revised the manuscript.

[Figure]

Fig. 5. Hygroscopic growth factors of aerosol particles containing mixtures of 4-hydroxybenzoic acid with ammonium sulfate at three different dry mass ratios. The measurements and model calculations represent particle growth during hydration experiment from 5 % to 90 % RH at 298.15 K. Measured growth factors are corrected for the Kelvin effect and and therefore shown vs. water activity. Mass ratio of AS:4-Hydroxybenzoic Acid: **(a)** 3:1, **(b)** 1:1, **(c)** 1:3.

**Page 14 line 377-379, we add:** "For the case of these mixtures during moistening, the deliquescence of ammonium sulfate in the mixed particles remains unaffected, within experimental resolution, by the presence of 4-hydroxybenzoic acid (Lei et al., 2014). Similar behavior has been observed for particles containing certain organic acids of limited water-solubility mixed ammonium sulfate (Choi and Chan, 2002; Chan and Chan, 2003). For example, mixtures for succinic acid + ammonium sulfate showed no substantial influence on the deliquescence RH of ammonium sulfate in the hydration process (Choi and Chan, 2002). However, a clear RH shift of the deliquescence phase transition of ammonium sulfate or sodium chloride was determined for mixed particles containing organic acids of higher water-solubility and O: C ratio, such as citric acid and malonic acid (e.g. Choi and Chan, 2002)."

*12. Page 15: They authors attempt to rationalize Figure 4 by two-liquid-to-solid phase transition. Subsequent discussion uses the term "liquid phase". What liquid phase do the authors refer? Organic or aqueous. I cautioned that without additional information, it is rather speculative to attempt to explain the behavior of the mixed particles, especially that the difference between the experimental data and either model predictions is not that large.*

**Response:** "The liquid phase" that we refer to is further clarified in the revised text below. We agree with the reviewer that the difference between the experimental data and either model predications is not large, but interesting and noticeable at the higher organic mass fractions in the mixtures. We indicated the speculative nature of our proposed explanation on page 14, line 386: "…, since the deviation from measurements might partly be explained by a transition in the physical state of the organic component." and on line 393. We further stress this caveat in the revised discussion. It is also worth noting that according to the AIOMFAC model predictions, the difference due to a liquid-solid phase transition would not be expected to be large. In addition, we mention in the revised manuscript that the AIOMFAC model predicts a liquid-liquid phase separation (LLPS) to exists below ~90 % RH for the calculation cases with the assumption of the organic in the liquid state (for all three organic mass fractions). This LLPS is responsible for the kink in the red AIOMFAC curve near 90 % RH in Fig. 4.

**Related changes included in the revised manuscript:**

**Page 15 line 403-407, we revise the statements starting with "Meanwhile,…." was revised to:**
"Similarly, a liquid-to-solid phase transition occurs for the organic: AS mass ratio of 3:1 cases in the range from 80 % to 50 % RH. Meanwhile, AS remains dissolved in a supersaturated aqueous solution phase. Moreover, the AIOMFAC-based equilibrium model predicts a liquid-liquid phase separation (LLPS) to occur at RH below ~90 % for the calculation cases with the assumption of the organic in the liquid state (for all three organic mass fractions in Fig. 4). This prediction leads to a liquid phase enriched in 4-hydroxybenzoic acid with some water and AS dissolved and a coexisting liquid phase enriched in AS and water. The onset of the LLPS during dehumidification leads to the kink in the red model curve near 90 % RH, since the slope of the HGF curve with RH changes in a non-smooth manner at the point of the LLPS phase transition. This change in slope is not noticeable from the experimental data alone, but the model-measurement comparison for the range above 80 % RH shows very good agreement. The two liquid phases will likely remain separated until nucleation of a crystalline 4-hydroxybenzoic acid phase occurs followed by gradual

partitioning of the organic acid to the solid phase with decreasing RH (to ~50 % RH), at which point only a single liquid phase (an aqueous AS phase with a tiny amount of dissolved HA) will remain until efflorescence of AS occurs. Above ~90 % RH, a single, homogeneous liquid phase is predicted to exist. Interestingly, this AIOMFAC model-measurement comparison (Fig. 4, especially panels b and c) provides reasonable evidence that 4-hydroxybenzoic acid remains dissolved and therefore in a liquid phase state at high RH in the mixed particles upon dehumidification (it is present in both liquid phases below 90 % RH, but highly enriched in the AS-poor phase)."

**Add caveat on page 15 line 398, we add after "with 50 and 75 wt % organic":**

"We acknowledge that the model predictions of the HGF curves for the two organic phase state assumptions differ within experimental error for the case shown in Fig. 4a, indicating that alternative explanations, such as model/measurement uncertainty in the absence of a liquid-solid phase transition could explain the observations."

*13. 3.3.1 Give more explanation on why the chosen mixtures would represent Amazon BBOA in dry and wet seasons.*

**Response:** According to the referee's suggestions, we add some additional discussion in the revised manuscript on the organic surrogate compounds chosen for representing biomass burning particles.

**Related additions and changes included in the revised manuscript:**

**Page 16 line 434-443: "To represent of biomass burning organic compounds…..compounds and ammonium sulfate" was revised to**

"According to Decesari et al, (2006), sampling of aerosol particles, including the WSOC fraction, was conducted from September 9 to November 14, 2002 in their field study, the sampling time was subdivided into different periods. Despite of significant changes in the chemical composition of tracer compounds from the dry to the wet period, the functional groups and general chemical classes of WSOC changed only to a small extent in the Amazon basin near Rondônia, Brazil. Model compounds represent semi-quantitatively (presence/abundance of functional groups) and the chemical structure of WSOC can be used as surrogates in microphysical models involving organic aerosol particles over tropical areas affected by biomass burning scenarios (Andreae et al., 2002; Artaxo et al., 2002; Rissler et al., 2006; Decesari et al., 2006). Here, we focus on experimental observations and model calculations for relatively simple mixtures of inorganic-organic surrogate components reflecting mixtures of aerosol components found during different seasons during

biomass burning events. However, we are fully aware of that fact that actual biomass burning aerosols are typically much more complex in terms of particle chemical composition. Aerosol particle properties from biomass burning events depend on the types of sources, external/internal population mixing state, water-solubilities, and phase state of the diversity of organic compounds and their mixing with inorganic constituents during different time periods in the field (e.g. Decesari et al., 2006)."

*14. Line 448-450. Cheung et al have examined GF of ambient atmospheric particles as a function of RH. They conclude that Kappa at high RH (like 90%) can be different from those at 50% or below. The authors can elaborate more on whether their observation is general for ambient measurements or just a characteristic of BBOA/AS.*

**Response:** As the referee points out, Cheung et al. (2015) have studied the hygroscopicity parameter $\kappa$ of ambient particles in Hong Kong using a HTDMA. They typically observed an increase in the hygroscopicity parameter $\kappa$ with decreasing RH from 90 % to 40 % RH. A similar trend is also observed in other field measurements. For example, Rastak et al. (2017) discuss the $\kappa$ parameters obtained measured for SOA produced from the photooxidation of monoterpenes or isoprene at subsaturated conditions. In agreement with Cheung et al. (2015), it is also found that $\kappa$ can vary substantially in the RH range from 50 to 90 %. The variability of the hygroscopicity parameter $\kappa$ in sub-saturated conditions reveals limitations of a single-parameter hygroscopicity model for applications over a wide range of RH. at low, intermediate and high RH levels, differing degrees of solution non-ideality, potential for liquid-liquid phase separation, water-solubility limitations of organics in ambient organic-inorganic particles, and assumptions about constant/variable surface tension may all play a role (Rastak et al., 2017; Ovadnevaite et al., 2017; Wang et al., 2017). An increase of the $\kappa$ value of SOA particles from the ozonolysis of α-pinene by a factor of 4-6 was observed in the RH-range from below 90 % to 99.6 % (Wex et al., 2009). They attributed the increase to non-ideal mixing effects in the solution droplet not fully accounted for by a single value of $\kappa$ and also to potential surface tension variation (while in the calculation of $\kappa$ from experimental data typically a constant surface tension value is assumed). Also, factors such as water content, dry particle size, and solubility have an influence on the computed/measured hygroscopicity parameter concentration-dependence of $\kappa$ (Mikhailov et al., 2009; Kim et al., 2016; Dawson et al., 2016; Wang et al., 2017). For example, a water activity dependence of $\kappa$ values for some aerosol particles has been shown for sub-saturation conditions by Mikhailov et al. (2009).

**Related additional and changes included in the revised manuscript:**

**Page 17 line 448-450, we added:** "Hygroscopicity parameter values for bio-mix-dry aerosol particles were determined to be between 0.16 and 0.18 with decreasing RH in the range from 90 % to 40 % RH. The $\kappa$ value representing the wet period in the Amazon is shown in Fig. 6b, derived from laboratory HTDMA measurements in the range from 90 % to 40 % RH. A similar trend of an increase in $\kappa$ with a decrease in RH has also been observed by Cheung et al. (2015). Their observation is based on ambient particle measurement with a HTDMA in Hong Kong, therefore probing particles of more complex compositions in the field campaign. The variability of the hygroscopicity parameter in sub-saturated conditions reveals some limitations of a single-parameter hygroscopicity model for applications over a wide range of RH. At low, intermediate and high RH levels, differing degrees of solution non-ideality, potential for liquid-liquid phase separation, water-solubility limitations of organics in ambient organic-inorganic particles, and assumptions about constant/variable surface tension may all play a role (Mikhailov et al. 2009; Wex et al., 2009, Rastak et al., 2017; Ovadnevaite et al. 2017; Wang et al., 2017)."

*15. The difference between Kappa based on subsaturated and supersaturated measurements can be due to sparingly soluble materials. See Huff Hartz et al (2006) and Chan et al. (2008) for more details. Does 4-hydroxybenzoic acid represent a sparingly soluble organic?*

**Response**: Yes. As referee suggested, Huff Hartz et al. (2006) and Chan et al. (2008) measured CCN activation of different organic species with water solubilities ranging from 0.0001 to 100 g solute $100g^{-1}$ $H_2O$. They found that certain organic compounds of very low solubility (0.3g-1g per 100g $H_2O$) can still exhibit considerable CCN activity at typical supersaturation. 4-hydroxybenzoic acid is considered as moderate soluble compounds with a solubility of (0.6751 g/100 g) (Lei et al., 2014). In particular for CCN activity predictions vs. hygroscopicity at subsaturation, it needs to be considered that $\kappa$ parameters are not constant.

**Related additions and changes included in the revised manuscript:**

**Page 18 line 481-485, we add:** "For example, the $\kappa$ parameter obtained from field data is ~0.15±0.06 at 90 % RH, while its value reaches ~0.18±0.04 at RH > 100 % (just prior to CCN activation). A likely reason for the difference is that hygroscopic particles, especially those containing sparingly soluble organics like 4-hydroxybenzoic acid, take up water dramatically above 95 % RH when approaching 100 % RH (Huff Hartz et al., 2006; Chan et al., 2008; Rastak et al., 2017), which is clear from model predictions, as demonstrated in Fig. 6 by application of Eq.

(4). The predicted curve in the mixture's effective $\kappa$ parameter may well capture the change in hygroscopicity under such high RH conditions."

**On page 18, line 485, we add:** "Consequently, for a precise representation of the hygroscopic growth behavior (e.g. HGF) at high RH (> 95 %) by the $\kappa$-Köhler model, the value of $\kappa$ would need to be varied. While a variable $\kappa$ value is contrary to the attempted simplicity of the single-parameter $\kappa$-Köhler model, it is at least advised to consider that $\kappa$ values derived from HGF data at 80 % or 90 % RH may not apply accurately for the calculation of CCN activation properties of such biomass burning aerosols."

**References:**

Brooks, S. D., DeMott, P. J., and Kreidenweis, S. M.: Water uptake by particles containing humic materials and mixtures of humic materials with ammonium sulfate, Atmos. Environ., 38, 1859-1868, 2004.

Chan, M. N. and Chan, C. K.: Hygroscopic properties of two model humic-like substances and their mixtures with inorganics of atmospheric importance, Environmental Science & Technology, 37, 5109-5115, 2003.

Chan, M. N., Choi, M. Y., Ng, N. L., and Chan, C. K.: Hygroscopicity of water-soluble organic compounds in atmospheric aerosols: Amino acids and biomass burning derived organic species, Environmental Science & Technology, 39, 1555-1562, 2005.

Chan, M. N., Kreidenweis, S. M., and Chan, C. K.: Measurements of the Hygroscopic and Deliquescence Properties of Organic Compounds of Different Solubilities in Water and Their Relationship with Cloud Condensation Nuclei Activities, Environmental Science & Technology, 42, 3602-3608, 2008.

Chan, M. N. C. a. C. K.: Mass transfer effects in hygroscopic measurements of aerosol particles, Atmospheric Chemistry and Physics, 2005.

Cheung, H. H. Y., Yeung, M. C., Li, Y. J., Lee, B. P., and Chan, C. K.: Relative Humidity-Dependent HTDMA Measurements of Ambient Aerosols at the HKUST Supersite in Hong Kong, China, Aerosol Science and Technology, 49, 643-654, 2015.

Choi, M. Y. and Chan, C. K.: The effects of organic species on the hygroscopic behaviors of inorganic aerosols, Environmental science & technology, 36, 2422-2428, 2002.

Ciobanu, V. G., Marcolli, C., Krieger, U. K., Zuend, A., and Peter, T.: Efflorescence of ammonium sulfate and coated ammonium sulfate particles: evidence for surface nucleation, The Journal of Physical Chemistry A, 114, 9486-9495, 2010.

Claeys, M., Kourtchev, I., Pashynska, V., Vas, G., Vermeylen, R., Wang, W., Cafmeyer, J., Chi, X., Artaxo, P., Andreae, M. O., and Maenhaut, W.: Polar organic marker compounds in atmospheric aerosols during the LBA-SMOCC 2002 biomass burning experiment in Rondônia, Brazil: sources and source processes, time series, diel variations and size distributions, Atmospheric Chemistry and Physics, 10, 9319-9331, 2010.

Decesari, S., Fuzzi, S., Facchini, M. C., Mircea, M., Emblico, L., Cavalli, F., Maenhaut, W., Chi, X., Schkolnik, G., Falkovich, A., Rudich, Y., Claeys, M., Pashynska, V., Vas, G., Kourtchev, I., Vermeylen, R., Hoffer, A., Andreae, M. O., Tagliavini, E., Moretti, F., and Artaxo, P.: Characterization of the organic composition of aerosols from Rondonia, Brazil, during the LBA-SMOCC 2002 experiment and its representation through model compounds, Atmospheric Chemistry and Physics, 6, 375-402, 2006.

Dusek, U., Frank, G. P., Massling, A., Zeromskiene, K., Iinuma, Y., Schmid, O., Helas, G., Hennig, T., Wiedensohler, A., and Andreae, M. O.: Water uptake by biomass burning aerosol at sub- and supersaturated conditions: closure studies and implications for the role of organics, Atmospheric Chemistry and Physics, 11, 9519-9532, 2011.

Engling, G., Lee, J. J., Sie, H.-J., Wu, Y.-C., and I, Y.-P.: Anhydrosugar characteristics in biomass smoke aerosol—case study of environmental influence on particle-size of rice straw burning aerosol, Journal of Aerosol Science, 56, 2-14, 2013.

Gysel, M., Weingartner, E., Nyeki, S., Paulsen, D., Baltensperger, U., Galambos, I., and Kiss, G.: Hygroscopic properties of water-soluble matter and humic-like organics in atmospheric fine aerosol, Atmospheric Chemistry and Physics, 4, 35-50, 2004.

Hanford, K. L., Mitchem, L., Reid, J. P., Clegg, S. L., Topping, D. O., and McFiggans, G. B.: Comparative thermodynamic studies of aqueous glutaric acid, ammonium sulfate and sodium chloride aerosol at high humidity, Journal of Physical Chemistry A, 112, 9413-9422, 2008.

Hodas Natasha, Z. A., Schilling Katherine, Berkemeier Thomas, Shiraiwa Manabu, Flagan Richard C, and Seinfeld John H.: Discontinuities in hygroscopic growth below and above water saturation for laboratory surrogates of oligomers in organic atmospheric aerosols, Atmos. Chem. Phys., doi: 10.5194/acp-16-12767-2016, 2016.

Hoffer, A., Gelencser, A., Guyon, P., Kiss, G., Schmid, O., Frank, G. P., Artaxo, P., and Andreae, M. O.: Optical properties of humic-like substances (HULIS) in biomass-burning aerosols, Atmospheric Chemistry and Physics, 6, 3563-3570, 2006.

Huff Hartz, K. E., Tischuk, J. E., Chan, M. N., Chan, C. K., Donahue, N. M., and Pandis, S. N.: Cloud condensation nuclei activation of limited solubility organic aerosol, Atmos. Environ., 40, 605-617, 2006.

Iinuma, Y., Brüggemann, E., Gnauk, T., Müller, K., Andreae, M. O., Helas, G., Parmar, R., and Herrmann, H.: Source characterization of biomass burning particles: The combustion of selected European conifers, African hardwood, savanna grass, and German and Indonesian peat, Journal of Geophysical Research, 112, 2007.

Jing, B., Tong, S., Liu, Q., Li, K., Wang, W., Zhang, Y., and Ge, M.: Hygroscopic behavior of multicomponent organic aerosols and their internal mixtures with ammonium sulfate, Atmospheric Chemistry and Physics, 16, 4101-4118, 2016.

Kerminen, V. M.: The effects of particle chemical character and atmospheric processes on particle hygroscopic properties, Journal of Aerosol Science, 28, 121-132, 1997.

Lei, T., Zuend, A., Wang, W. G., Zhang, Y. H., and Ge, M. F.: Hygroscopicity of organic compounds from biomass burning and their influence on the water uptake of mixed organic ammonium sulfate aerosols, Atmospheric Chemistry and Physics, 14, 1-20, 2014.

Liu, Q., Jing, B., Peng, C., Tong, S., Wang, W., and Ge, M.: Hygroscopicity of internally mixed multi-component aerosol particles of atmospheric relevance, Atmos. Environ., 125, 69-77, 2016. Mikhailov, E., Vlasenko, S., Rose, D., and Pöschl, U.: Mass-based hygroscopicity parameter interaction model and measurement of atmospheric aerosol water uptake, Atmospheric Chemistry and Physics, 13, 717-740, 2013.

Mikhailov, E. F., Mironov, G. N., Pöhlker, C., Chi, X., Krüger, M. L., Shiraiwa, M., Förster, J. D., Pöschl, U., Vlasenko, S. S., Ryshkevich, T. I., Weigand, M., Kilcoyne, A. L. D., and Andreae, M. O.: Chemical composition, microstructure, and hygroscopic properties of aerosol particles at the Zotino Tall Tower Observatory (ZOTTO), Siberia, during a summer campaign, Atmos. Chem. Phys., 15, 8847-8869, 2015.

Mochida, M. and Kawamura, K.: Hygroscopic properties of levoglucosan and related organic compounds characteristic to biomass burning aerosol particles, Journal of Geophysical Research-Atmospheres, 109, 2004.

Ovadnevaite, J., Zuend, A., Laaksonen, A., Sanchez, K. J., Roberts, G., Ceburnis, D., Decesari, S., Rinaldi, M., Hodas, N., Facchini, M. C., Seinfeld, J. H., and O' Dowd, C.: Surface tension prevails over solute effect in organic-influenced cloud droplet activation, Nature, 546, 637-641, 10.1038/nature22806, 2017.

Peng, C., Chan, M. N., and Chan, C. K.: The hygroscopic properties of dicarboxylic and multifunctional acids: Measurements and UNIFAC predictions, Environmental Science & Technology, 35, 4495-4501, 2001a.

Peng, C., Chan, M. N., and Chan, C. K.: The Hygroscopic Properties of Dicarboxylic and Multifunctional Acids: Measurements and UNIFAC Predictions, Environmental Science & Technology, 35, 4495-4501, 2001b.

Petters, M. D. a. K., S. M.: A single parameter representation of hygroscopic growth and cloud condensation nucleus activity, Atmos. Chem. Phys., 2008.

Samburova, V., Hallar, A. G., Mazzoleni, L. R., Saranjampour, P., Lowenthal, D., Kohl, S. D., and Zielinska, B.: Composition of water-soluble organic carbon in non-urban atmospheric aerosol collected at the Storm Peak Laboratory, Environmental Chemistry, 10, 370, 2013.

Smith, M. L., Bertram, A. K., and Martin, S. T.: Deliquescence, efflorescence, and phase miscibility of mixed particles of ammonium sulfate and isoprene-derived secondary organic material, Atmospheric Chemistry and Physics, 12, 9613-9628, 2012.

Wang, Z., Cheng, Y., Ma, N., Mikhailov, E., Pöschl, U., and Su, H.: Dependence of the hygroscopicity parameter κ on particle size, humidity and solute concentration: implications for laboratory experiments, field measurements and model studies, Atmos. Chem. Phys. Discuss., 2017, 1-33, 2017.

Wex, H., Petters, M. D., Carrico, C. M., Hallbauer, E., Massling, A., McMeeking, G. R., Poulain, L., Wu, Z., Kreidenweis, S. M., and Stratmann, F.: Towards closing the gap between hygroscopic growth and activation for secondary organic aerosol: Part 1 – Evidence from measurements, Atmos. Chem. Phys., 9, 3987-3997, 2009.

Whitehead, J. D., Darbyshire, E., Brito, J., Barbosa, H. M., Crawford, I., Stern, R., Gallagher, M. W., Kaye, P. H., Allan, J. D., and Coe, H.: Biogenic cloud nuclei in the central Amazon during the transition from wet to dry season, Atmospheric Chemistry and Physics, 16, 9727-9743, 2016.

Zamora, I. R., Tabazadeh, A., Golden, D. M., and Jacobson, M. Z.: Hygroscopic growth of common organic aerosol solutes, including humic substances, as derived from water activity measurements, Journal of Geophysical Research: Atmospheres, 116, n/a-n/a, 2011.

Zardini, A. A., Sjogren, S., Marcolli, C., Krieger, U. K., Gysel, M., Weingartner, E., Baltensperger, U., and Peter, T.: A combined particle trap/HTDMA hygroscopicity study of mixed inorganic/organic aerosol particles, Atmospheric Chemistry and Physics, 8, 5589-5601, 2008.

**Authors' response to comments by anonymous referee #2:**

*This study presents an attempt to distinguish the role of particle phase state on the hygroscopicity of biomass burning surrogates and mixtures with ammonium sulfate. I consider the addition of such studies to the literature worthwhile. However, this paper requires a number of changes and clarifications before being accepted for publication. Before these are clarified, I found it difficult to provide further critique on a number of results presented. After reading the first review, which I tend to agree with on specific points raised, I present a number of different factors the authors need to address below:*

**Response:** We are grateful to referee #2 for her/his comments and suggestions to improve our manuscript. We have implemented changes based on these precious specific comments in the revised manuscript of the article. We repeat the specific points raised by the reviewer in italic font, followed by our response. The pages numbers and lines mentioned are with respect to the Atmospheric Chemistry and Physics Discussions (ACPD) paper (original version).

Specific comments and author response:

*(1): Abstract: I would recommend removing reliance on the word 'slightly'. Please quantify 'slightly' or remove entirely. This paper often feels a little too qualitative in nature, and I would recommend checking all instances like this. The authors also comment on 'similarity of hygroscopic parameter k'. Please quantify this. What range do you consider to be similar? line 35: Presume the authors mean sub-saturated 'RH'. line 35: 'at' the same environment? This is unclear. I would recommend checking all grammar throughout the document, as also noted by the other referee.*

**Reply:** As this referee pointed out some grammar deficiencies, we rephrased several statements to improve grammar, wording and sentence structure. Further rephrasing will also be considered in preparation of the revised version of this article.

First, the word "slightly" has the meaning "very small in degree or amount", which is not quantitative, but implies a small change relative to the magnitude of the quantity it is referring to.

As such, it is not untypical to be found it in scientific literature. However, we agree that the sentences concerned can be improved by better wording.

Second, concerning the statement "A similarity of the hygroscopicity parameter $\kappa$ for organic surrogate compounds mixed with ammonium sulfate for different mass fractions during the different seasonal periods in the Amazon is observed.", figure 6a shows a small difference in the hygroscopicity parameter $\kappa$ for mixtures of organic surrogate compounds and ammonium sulfate representing biomass burning particles during the dry and wet periods in the central Amazon Basin. For example, measured $\kappa$ values for bio-mix-dry aerosol particles were determined to be between 0.16 and 0.18 in the range from 90 to 40 % RH using the HTDMA technique, which is slightly higher than that of $\kappa$ determined for the bio-mix-wet aerosol particles ($\kappa \approx 0.12$ - 0.15). We have revised these sentences in the manuscript.

Third, "the RH-dependent hygroscopicity parameter $\kappa$ for organic surrogate compounds….at the same environment,….field observation conditions" at the same environment refer to the RH condition, Here, hygroscopicity parameter $\kappa$ at 90 % RH in the laboratory compared with Kappa $\kappa$ at the same RH.

**Related additions and changes included in the revised manuscript were made for the following sentences:**
**Page 1 line 17:** "We observed that levoglucosan and humic acid aerosol particles release water from upon dehumidification in the range from 90 % − 5 % relative humidity (RH)."

**Page 1 line 18:** "4-Hydroxybenzoic acid aerosol particles, however, remain in the solid state without diameter growth both upon humidification or dehumidification and exhibit a small shrinking in size at higher RH compared to the dry size. For example, the measured growth factor of 4-hydroxybenzoic acid aerosol particles is ~0.96 at 90 % RH."

**Page 13 line 345:** "The E-AIM prediction is in relatively good agreement with results from the HTDMA measurement but typically overestimates the water content of particles consisting of mixtures at the RH range close to the ERH of AS."

**Page 2 line 32-37, we revise**: "Lastly, two distinct mixtures of organic surrogate compounds, including levoglucosan, 4-hydroxybenzoic acid, and humic acid were used to represent the average water-soluble organic carbon (WSOC) fractions observed during the wet and dry seasons in central Amazon Basin. A comparison of the organic hygroscopicity parameter for the simple mixtures, e.g.

$\kappa \approx 0.12$ to 0.15 for the wet-season mixture in the 90 % to 40 % RH range, shows good agreement with field data for the wet season in Amazon (WSOC $\kappa \approx 0.14 \pm 0.06$ at 90 % RH). This suggests that laboratory-generated mixtures containing organic surrogate compounds and ammonium sulfate can be used to mimic, in a simplified manner, the chemical composition of ambient aerosols from the Amazon for the purpose of RH-dependent hygroscopicity studies."

**Page 5 line 124-126, we add:** "The three organic compounds levoglucosan, 4-hydroxybenozic acid and humic acid were used as surrogates for the rich-class of water-soluble organic components in biomass burning aerosols. The influence of the distinct chemical structure of these compounds was studied with regard to the water uptake and evaporation of the pure organic compounds as well as for mixed organic-AS-containing particles. Furthermore, a comparison with field data from the Amazon was preformed to quantify the ability of mixtures of these three organic compounds to mimicking the hygroscopic behavior of complex ambient organic particles originating from biomass burning emissions."

**Page 10 line 284-287, we add:** "Also, no ERH of 4-hydroxybenzoic acid in the dehydration mode was observed during the experiments; the likely reason is that the highest RH reached in the humidifier was approximately 98 %, which may be below the ERH of 4-hydroxybenzoic acid, reported as possibly above 95 % RH in another study (Mochida and Kawamura, 2004)."

*(2) Page 7, line 176: 'Here the AIOMFAC-based thermodynamic equilibrium model is used to calculate the DRH...in the multicomponent system based on the known solubility of AS in the organic-free system.? I'm not sure why you have chosen to do this when the benefit of the AIOMFAC activity coefficient model is to account for inorganic-organic interactions. Please justify this as the proceeding equations do not necessarily correlate with this statement.*

**Reply:** The reviewer is right, that an advantage of the AIOMFAC-based model is to account for inorganic-organic interactions in liquid mixtures. Such interactions will change the activity coefficient and thus we determined ERH and DRH not based on the soluteconcentration but the activities calculated by AIOMFAC. The given statements in the text are correct and consistent with Eq. (1); however, the sentence is perhaps not clear enough in the given form. We have therefore made a few revisions to clarify that knowledge of the solubility of AS in the organic-free system can be used to calculate its solubility in organic-inorganic mixtures with the help of a thermodynamic model.

**Related additions and changes included in the revised manuscript:**

**Page 7 line 169-178, we add**: "For example, in the case of a ternary liquid mixture of levoglucosan + AS + water in solid-liquid equilibrium (SLE) with a crystalline AS phase at a certain temperature $T$, a constant molal ion activity product $\text{IAP}_{\text{AS}} = \text{IAP}_{\text{AS}}^{(\text{sat})}(T)$ is established (necessary SLE condition). In this case the liquid mixture is a so-called saturated solution with respect to AS. While the molar amount of AS in a saturated solution depends on the other mixture constituents, the value of $\text{IAP}_{\text{AS}}^{(\text{sat})}(T)$ is a function of temperature only, since it is derived from the fixed chemical composition and associated chemical potential of the crystalline phase. A reference value for $\text{IAP}_{\text{AS}}^{(\text{sat})}(T)$ can therefore be calculated with the AIOMFAC model from an experimentally determined solubility limit of AS in a known mixture, such as the molality of AS at the point of saturation in the binary aqueous system (water + AS). The RH at which full dissolution of a solid phase upon humidification is just reached, the DRH, is directly related to the conditions at which a saturated solution becomes subsaturated upon addition of water. Here the degree of saturation with AS can be determined unambiguously by the computed value of $\text{IAP}_{\text{AS}}$ as a function of mixture composition and temperature. Making use of these thermodynamic relationships, the AIOMFAC-based equilibrium model is used to calculate the DRH and ERH of AS in the multicomponent system, as outlined below."

**Page 7 line 188, we add**: "The RH at which this $\text{IAP}_{\text{AS}}^{(\text{sat})}$ value is just reached in a certain bulk solution at equilibrium with its environment (in contrast to $\text{IAP}_{\text{AS}} < \text{IAP}_{\text{AS}}^{(\text{sat})}$ at higher RH), is the (bulk) DRH of AS."

*(3) Page 8: line 209 'Differences in the density models are expected to lead to relatively small differences.' This needs a qualifying reference or a demonstration of sensitivity. What do you mean by relatively small?*

**Reply:** Fig. 1 shows the impact of different density treatment in the E-AIM and the AIOMFAC-based models. In the case of predicted mass growth factors of ammonium sulfate, both models agree well with each other, especially at RH > 70 % RH, indicating that the corresponding differences in predicted diameter growth factors must be due to the different way the conversion

from particle mass to particle volume is done by the two models. The expected difference is on the order of the HTDMA measurement error or less.

[Figure]

Fig. 1. Comparison of E-AIM and AIOMFAC-based mass growth factors (a) and growth diameter growth factors (b) for the binary ammonium sulfate + water system at 298.15 K (Fig. 4 of Lei et al., 2014).

**Related additions and changes included in the revised manuscript:**

**Manuscript revision. We revise the corresponding sentence (page 8, line 209) to include the reference:** "Differences in the density models are expected to lead to relatively small difference, typically on the order of the HTDMA measurement error or less (e.g. Fig. 2a), in the application to HGF predictions, as demonstrated by Lei et al. (2014) for the case of diameter vs. mass-based HGF of AS droplets."

*(4) Page 10, line 272: 'standard UNIFAC..'. Which set of interaction parameters are you using? A reader should be able to replicate these results.*

**Reply:** We use the set of UNIFAC interaction parameters included by default in the E-AIM model and in AIOMFAC. To clarify a bit further, we add an additional statement in the manuscript.

**Related additions and changes included in the revised manuscript:**

**On page 10, line 273, we add**: "The UNIFAC models within E-AIM and AIOMFAC are based on the original model expressions by Fredenslund et al. (1975) and both include the extensive parameter set by Hansen et al. (1991) as well as revised parameters for certain group interactions of water with carboxyl and hydroxyl groups by Peng et al. (2001). Of relevance for levoglucosan and other sugar-like compounds, the AIOMFAC model also contains certain revised group parameters for hydroxyl groups and special alkyl groups for their interactions with water, introduced by Marcolli and Peter (2005) for polyols, as further detailed in Zuend et al. (2011)."

*(5) Page 10, line 276:'intra molecular interactions are fully considered by these models..'. What is UNIFAC based on?*

**Reply:** The referee probably misread the sentence. The sentence in the manuscript reads: "However, the molecular structure of levoglucosan with several polar functional groups in close vicinity may account for a small deviation between models and measured HGFs at RH below 70 %, because intramolecular interactions are **not** fully considered by these models."

UNIFAC is an abbreviation that stands for Universal quasichemical Functional Group Activity Coefficients. It is based on the quasichemical theory of liquid solution s proposed by Guggenheim (1950), generalized by Abrams and Prausnitz (1975), and applied to functional group within the molecules (group-contribution concept) by Fredenslund et al. (1975). The UNIFAC method allows to calculate/predicts liquid-phase activity coefficients of non-electrolyte solutions.

*(6) Page 18, line 482:'at RH > 95% the water content of hygroscopic particles increases dramatically with a small increase in RH, leading to the predicted change in the mixtures k parameter that is best representing the hygroscopic growth under such high-RH conditions'. This statement is not clear. Are you suggesting that variable 'k' values are required? Please rephrase and clarify.*

**Reply:** Yes, if it is attempted to represent the hygroscopic behavior at high RH accurately, in particular for CCN activity predictions vs. hygroscopicity at subsaturation - it needs to be considered that $\kappa$ parameters are not constant and their values depend on the RH range of measurements or models from which they were determined; see, e.g. the discussion by Rastak et

al. (2017) or Wang et al. (2017). The field data from the Amazon as well as the prediction with Eq. (4) based on the laboratory measurements indicate a marked difference in the determined appropriate hygroscopicity parameter $\kappa$ between subsaturated and supersaturated water vapor conditions, with a transition predominantly in the RH from 90 % RH to ~100 % (Rastak et al., 2017). For example, the $\kappa$ parameter obtained from field data is ~0.15±0.06 at 90 % RH, while its value reaches ~0.18±0.04 at RH > 100 % (just prior to CCN activation). The reason for the difference is that hygroscopic particles uptake water dramatically above 95 % RH when approaching 100 % RH, which is clear from model predictions, as demonstrated in Fig. 6 by application of Eq. (4).

**Related additions and changes included in the revised manuscript:**

**Page 18 line 481-485, we add:** "For example, the $\kappa$ parameter obtained from field data is ~0.15±0.06 at 90 % RH, while its value reaches ~0.18±0.04 at RH > 100 % (just prior to CCN activation). A likely reason for the difference is that hygroscopic particles, especially those containing sparingly soluble organics like 4-hydroxybenzoic acid, take up water dramatically above 95 % RH when approaching 100 % RH (Huff Hartz et al., 2006; Chan et al., 2008; Rastak et al., 2017), which is clear from model predictions, as demonstrated in Fig. 6 by application of Eq. (4). The predicted curve in the mixture's effective $\kappa$ parameter may well capture the change in hygroscopicity under such high RH conditions."

**On page 18, line 485, we add:** "Consequently, for a precise representation of the hygroscopic growth behavior (e.g. HGF) at high RH (> 95 %) by the $\kappa$-Köhler model, the value of $\kappa$ would need to be varied. While a variable $\kappa$ value is contrary to the attempted simplicity of the single-parameter $\kappa$-Köhler model, it is at least advised to consider that $\kappa$ values derived from HGF data at 80 % or 90 % RH may not apply accurately for the calculation of CCN activation properties of such biomass burning aerosols."

*General comments:*

*(1) Please add a brief discussion on the expected performance of activity models for 4-hydroxybenzoic acid. It would help the reader understand where sensitivities might lie.*

**Replay:** According to the reviewer's suggestion, we have added some additional discussion to the revised manuscript to highlight that the difference between models and measurements are not due to model error, but due to assumptions about the physical state, as mentioned on page 11, line 293.

**Related additions and changes included in the revised manuscript:**

**On page 11, line 291, we add the following statements after "obvious in Fig. 2c":** "These deviations surpass by far the expected error in model performance, which is typically less than 0.05 in HGF units for RH < 85 %, as indicated also by an intercomparison of the AIOMFAC and E-AIM predictions in Fig. 2c and the much improved model-measurement agreement for the case of mixed 4-hydroxybenzoic acid + AS particles shown in Fig. 4 (discussed in Section 3.2.2). However, note that the validity of the shown model predictions in Fig. 2c depends on whether the assumption of a liquid solution droplet is plausible."

*(2) Have the authors considered how a variable morphology would influence results? Is there no literature data on studies using AIOMFAC on this?*

Response: We add some discussion on potential particle morphology effects. To our knowledge, there are no studies where morphology (non-sphericity) of solid particles is explicitly considered with AIOMFAC. In the case of phase-separated particles, overall sphericity is still assumed in AIOMFAC-based HGF predictions.

**Related additions and changes included in the revised manuscript:**

**Page 11, line 296, we add:** "Morphology effects, such as the restructuring of non-spherical polycrystalline particles over a certain RH range or liquid-liquid phase-separated particles of non-spherical shapes, have been discussed by several groups (Sjogren et al., 2007; Reid et al., 2011; Lei et al., 2014). In the case of hygroscopic growth and deliquescence under hydration conditions for 4-hydroxybenzoic acid particles and mixtures of 4-hydroxybenzoic acid with ammonium sulfate. An offset between measurement and model predictions was observed both in the RH range below the deliquescence of the particles and above it, i.e. above 80 % RH, (Lei et al., 2014). It is suggested that deviations are primarily caused by a change in solid-state particle morphology during hydration, leading to a restructuring of the polycrystalline particle shape towards more compact, near-spherical shape as the RH approaches the particle deliquescence point. This would explain rather uncommon HGF values of less than 1.0 at elevated RH also shown in Fig. 2c. Similar

behavior was found for experimental growth factors of mixtures containing adipic acid and AS and systematic deviations between the associated ZSR predictions and observations by Sjogren et al. (2007). Thus, while experimental data hint to the possible influence of non-spherical particles and their humidity-induced restructuring as a source of uncertainty, model predictions of HGF, such as those with the AIOMFAC model, assume by default a spherical particle shape even for solid phases and/or in cases where LLPS is present."

*(3) What is the residence time of particles in the HTDMA? If there were a phase state change from which kinetic mass transfer limitations might apply, how might this change your conclusions?*

**Reply:** We add further discussion on the effect of potential mass transfer limitations for aqueous aerosol HGF measurements and for the solid-liquid phase transitions during deliquescence.

**Related additions and changes included in the revised manuscript:**
**Page 13 line 335-343, we add:** "the rather high viscosity of solutions containing levoglucosan is expected to increase considerably toward RH (Marshall et al., 2016). This increase in viscosity might impede the crystallization of AS in the mixed systems on the time scale of the experiment. Mass transfer limitation effects on the deliquescence process of crystalline organic particles and the water uptake or evaporation have been investigated in several experimental studies (Peng et al., 2001; Choi and Chan., 2002; Chan and Chan., 2005; Sjogren et al., 2007; Zardini et al., 2008; Ciobanu et al., 2010; Smith et al., 2012; Mikhailov et al., 2013; Hodas et al., 2015). Mass transfer limitations may impact the outcome of experiments significantly if the characteristic time scales for equilibration is similar to or larger than the residence time of particles in the experimental setup. In this study, the total residence time of the aerosol sample during the equilibration phase before entering the DMA2 is about 8 s. In order to improve the probability that the particles reach equilibrium with the target RH during this residence time, the monodisperse aerosol selected by DMA1 is first humidified to 98 % RH. The aerosol particles are then exposed to a lower target RH by a two-step process using double Nafion tubes. Kerminen (1997) estimated the necessary residence time for achievement of water equilibrium of aqueous droplets to be between $8 \times 10^{-6}$ s and 0.1 s for 100 nm and 500 nm particles, respectively. Therefore, the typical residence time of a few seconds in the humidification or dehumidification section in a HTDMA measurement is assumed to be sufficient for most equilibrium hygroscopicity measurements (Brooks et al., 2004;

Mikhailov et al., 2004). Moreover, our HGF results for the three pure organic components are in good agreement with respective data by Mochida and Kawamura, (2004), Brooks et al., (2004) and Chan and Chan (2005), conducted with different techniques and/or residence times. However, there are cases where water equilibration could be impeded substantially in the presence of highly viscous or glassy particles at low RH, e.g. for ternary sucrose + NaCl + water particles of > 6 µm in diameter studied by Bones et al. (2012), who report an equilibration time scale > 1000 s for such particles. Note that, aside from viscosity, there is an important size-dependence of the particles on the equilibration time scale (e.g. Koop et al. 2011). For aqueous 100 nm particles used in HTDMA experiments at room temperature, Bones et al. (2012) indicate that the equilibration time scale for water is likely only of concern for RH < 10 % in such an instrument. We therefore conclude that the residence time of 8 s is very likely sufficient to allow for equilibrium HGF measurements in dehydration mode, at least down to 10 % RH (when starting with aqueous solution droplets).

Mass transfer effects in hygroscopicity measurements of aerosol particles during hydration conditions have been encountered previously, particularly when a solid-liquid phase transition (deliquescence) is involved (Peng et al., 2001; Chan and Chan., 2005; Sjogren et al., 2006). For example, Peng et al. (2001) observed in electrodynamic balance (EDB) experiments under conditions of very slow humidification that glutaric acid aerosol particles showed a deliquescence phase transition in the RH range from 83 to 85 % over the course of several hours. This is a much longer time span than that of ~ 40 min for the deliquescence of other super-micron sized dicarboxylic acid particles (e.g., malonic acid) in EDB experiments. This observation indicates that the solid-liquid phase transition of glutaric acid particles may likely be mass transfer limited during the hydration process. In this context, it is possible that the deliquescence of initially solid, pure 4-hydroxybenzoic acid particles at RH > 97 % is further impeded by slow dissolution, which could have led to the absence of deliquesced particles (Fig. 2c) on experimental time scale.

*(4) I would appreciate more discussion on how the reliance on 3 organic surrogates influences conclusions for a SOA class that is likely much more complex. Are you studies sensitive to complexity, influenced by a discrete range of solubilities and 'step-like' transitions? How would you test this?*

**Reply:** According to the referee's suggestions, we will added more discussions on the reliance of organic surrogate compounds on hygroscopic behavior of mixtures of biomass burning in the revised manuscript.

In this study, we focus on two distinct aspects of hygroscopicity-related research: (1) well-defined hygroscopic growth factor measurements and modeling for simple organic-inorganic mixtures. (2) Comparison of the effective hygroscopicity parameter of relatively simple mixtures of representative organic biomass burning surrogates to hygroscopicity values ($\kappa$) obtained from field data. The experimental observations with well-defined organic-inorganic laboratory systems suggests that such systems exhibit a considerable variability with regard to liquid-liquid and solid-liquid phase transition during humidification/dehumidification. These phase transitions may be absent or different in other systems of highly complex biomass burning organic aerosols from the field. Moreover, whether typical field particles show step-like phase transitions cannot answered without direct measurements in the field or with carefully sampled ambient particles. Our experimental hygroscopicity data for model systems representing biomass burning particles tend to show that distinct step-like features in the hygroscopicity curves are at least less noticeable in the RH range from 90 to 40 % when several organics are mixed, which is in agreement with findings by Marcolli et al. (2004). Marcolli et al. (2004) suggest that mixtures containing many different organic compounds tend to thermodynamically favor the liquid state and suppress crystallization of individual organic compounds.

**Related additions and changes included in the revised manuscript:**
**Page 16 line 434-443: "To represent of biomass burning organic compounds…..compounds and ammonium sulfate" was revised to**
"According to Decesari et al, (2006), sampling of aerosol particles, including the WSOC fraction, was conducted from September 9 to November 14, 2002 in their field study, the sampling time was subdivided into different periods. Despite of significant changes in the chemical composition of tracer compounds from the dry to the wet period, the functional groups and general chemical classes of WSOC changed only to a small extent in the Amazon basin near Rondônia, Brazil. Model compounds represent semi-quantitatively (presence/abundance of functional groups) and the chemical structure of WSOC can be used as surrogates in microphysical models involving organic aerosol particles over tropical areas affected by biomass burning scenarios (Andreae et al., 2002; Artaxo et al., 2002; Rissler et al., 2006; Decesari et al., 2006). Here, we focus on experimental

observations and model calculations for relatively simple mixtures of inorganic-organic surrogate components reflecting mixtures of aerosol components found during different seasons during biomass burning events. However, we are fully aware of that fact that actual biomass burning aerosols are typically much more complex in terms of particle chemical composition. Aerosol particle properties from biomass burning events depend on the types of sources, external/internal population mixing state, water-solubilities, and phase state of the diversity of organic compounds and their mixing with inorganic constituents during different time periods in the field (e.g. Decesari et al., 2006)."

**Reference**

Bones, D. L., Reid, J. P., Lienhard, D. M., and Krieger, U. K.: Comparing the mechanism of water condensation and evaporation in glassy aerosol, Proceedings of the National Academy of Sciences, 109, 11613-11618, 10.1073/pnas.1200691109, 2012.

Brooks, S. D., DeMott, P. J., and Kreidenweis, S. M.: Water uptake by particles containing humic materials and mixtures of humic materials with ammonium sulfate, Atmos. Environ., 38, 1859-1868, 2004.

Chan, M. N. and Chan, C. K.: Hygroscopic properties of two model humic-like substances and their mixtures with inorganics of atmospheric importance, Environmental Science & Technology, 37, 5109-5115, 2003.

Chan, M. N. C. a. C. K.: Mass transfer effects in hygroscopic measurements of aerosol particles, Atmospheric Chemistry and Physics, 2005.

Decesari, S., Fuzzi, S., Facchini, M. C., Mircea, M., Emblico, L., Cavalli, F., Maenhaut, W., Chi, X., Schkolnik, G., Falkovich, A., Rudich, Y., Claeys, M., Pashynska, V., Vas, G., Kourtchev, I., Vermeylen, R., Hoffer, A., Andreae, M. O., Tagliavini, E., Moretti, F., and Artaxo, P.: Characterization of the organic composition of aerosols from Rondonia, Brazil, during the LBA-

SMOCC 2002 experiment and its representation through model compounds, Atmospheric Chemistry and Physics, 6, 375-402, 2006.

Gysel, M., Weingartner, E., Nyeki, S., Paulsen, D., Baltensperger, U., Galambos, I., and Kiss, G.: Hygroscopic properties of water-soluble matter and humic-like organics in atmospheric fine aerosol, Atmospheric Chemistry and Physics, 4, 35-50, 2004.

Marcolli, C. and Peter, Th.: Water activity in polyol/water systems: new UNIFAC parameterization, Atmos. Chem. Phys., 5, 1545-1555, doi:10.5194/acp-5-1545-2005, 2005.

Mochida, M. and Kawamura, K.: Hygroscopic properties of levoglucosan and related organic compounds characteristic to biomass burning aerosol particles, Journal of Geophysical Research-Atmospheres, 109, 2004.

Peng, C., Chan, M. N., and Chan, C. K.: The hygroscopic properties of dicarboxylic and multifunctional acids: Measurements and UNIFAC predictions, Environmental Science & Technology, 35, 4495-4501, 2001.

Kerminen, V. M.: The effects of particle chemical character and atmospheric processes on particle hygroscopic properties, Journal of Aerosol Science, 28, 121-132, 1997.

Koop, T., Bookhold, J., Shiraiwa, M., and Poschl, U.: Glass transition and phase state of organic compounds: dependency on molecular properties and implications for secondary organic aerosols in the atmosphere, Phys. Chem. Chem. Phys., 13, 19238-19255, 2011.

Lei, T., Zuend, A., Wang, W. G., Zhang, Y. H., and Ge, M. F.: Hygroscopicity of organic compounds from biomass burning and their influence on the water uptake of mixed organic ammonium sulfate aerosols, Atmospheric Chemistry and Physics, 14, 1-20, 2014.

---

## Author Response (AR2)

**Authors' response to comments by the editor:**

*Co-Editor Decision: Publish subject to minor revisions (review by editor) (28 Nov 2017) by*
*Markus Petters*

*Comments to the Author:*

*Dear authors,*

*I have received two second round reviews of your manuscript. Both referees are satisfied with the responses to the initial comments and I will be happy to accept the paper for publication. However, the manuscript must follow the ACP data policy (https://www.atmospheric-chemistry-and-physics.net/about/data_policy.html). To quote from this:*

*"Authors are required to provide a statement on how their underlying research data can be accessed. This must be placed as the section "Data availability" at the end of the manuscript before the acknowledgements. Please see the manuscript composition for the correct sequence. If the data are not publicly accessible, a detailed explanation of why this is the case is required."*

*In response to this item, please address in the manuscript how the data can be publicly accessed. Please note that "data will be made available upon request" is not acceptable under the ACP policy. The data should be freely available via a supplement to the manuscript or via a public archive.*

*Best regards,*

*Markus Petters*

**Reply**: Thank the editor for your information, we add some information as follows:

**Related additions and changes included in the revised manuscript:**

Page 24 line 666-667, we add: "**Data availability.** The measured results underlying this publication are available under the https://pan.baidu.com/s/1jIDxAcA."